# High-resolution ensemble projections and uncertainty assessment of regional climate change over China in CORDEX East Asia

Huanghe Gu[1,2], Zhongbo Yu[1,2], Chuanguo Yang[1,2], Qin Ju[1,2], Tao Yang[1,2], Dawei Zhang[3]

[1] State Key Laboratory of Hydrology-Water Resources and Hydraulic Engineering, Hohai University, Nanjing, China
[2] College of Hydrology and Water Resources, Hohai University, Nanjing, China
[3] China Institute of Water Resources and Hydropower Research, Beijing, China

*Correspondence to*: Zhongbo Yu (zyu@hhu.edu.cn)

**Abstract.** An ensemble simulation of five regional climate models (RCMs) from the coordinated regional downscaling experiment in East Asia is evaluated and used to project future regional climate change in China. The influences of model uncertainty and internal variability on projections are also identified. The RCMs simulate the historical (1980-2005) climate and future (2006-2049) climate projections under the Representative Concentration Pathway (RCP) RCP4.5 scenario. The simulations for five subregions in China, including Northeast China, North China, South China, Northwest China, and Tibetan Plateau, are highlighted in this study. Results show that (1) RCMs can capture the climatology, annual cycle and inter-annual variability of temperature and precipitation and a multi-model ensemble (MME) outperforms that of an individual RCM. The added values for RCMs are confirmed by comparing the performance of RCM and GCM in reproducing annual and seasonal mean precipitation and temperature during the historical period. (2) For future (2030-2049) climate, the MME indicates consistent warming trends at around 1 °C in the entire domain and projects pronounced warming in northern and western China. The annual precipitation is likely to increase in most of the simulation region, except for the Tibetan Plateau. (3) Generally, the future projected change in annual and seasonal mean temperature by RCMs is nearly consistent with the results from the driving GCM. However, changes in annual and seasonal mean precipitation exhibit significant inter-RCM difference and possess larger magnitude and variability than driving GCM. Even opposite signals for projected changes in average precipitation between the MME and the driving GCM are shown over South China, Northeast China and Tibetan Plateau. (4) The uncertainty in projected mean temperature mainly arises from the internal variability over north and south China and the model uncertainty over the rest three subregions. For the projected mean precipitation, the dominant uncertainty source is the internal variability over most regions, except for the Tibetan Plateau where the model uncertainty reaches up to 60%. Moreover, the model uncertainty increases with prediction lead time across all subregions.

## 1 Introduction

Globally averaged surface temperature increased by 0.65-1.06 °C from 1880 to 2012 according to several independently produced datasets, and further increases ranging from 0.3 °C to 4.8 °C are projected for 2081-2100 relative to 1986-2005 using a set of global climate models (GCMs) driven by the Representative Concentration Pathway (RCP) scenarios RCP2.6 to

RCP8.5 (IPCC, 2013). Meanwhile, other climate factors, such as precipitation amounts and variability, snow and ice cover patterns and mean sea level, are also changing (Alfieri et al., 2015; Kerr, 2008; Patz et al., 2005). Reliable projection of regional future climate is critical in evaluating climate change impacts and vulnerability and in developing appropriate mitigation and adaptation measures, especially for developing countries, such as China which tends to be one of the vulnerable countries to the adverse effects of climate changes (Kreft et al., 2016; Wang et al., 2017).

The East Asian summer monsoon (EASM) is the most distinctive climate feature in China, and the monsoon area accounts for approximately 60% of the mainland (Ding and Chan, 2005). EASM system-related precipitation starts around mid-May or even earlier in Indo-China Peninsula, which presents distinct stepwise northward and northeastward advances feature with two abrupt northward jumps and three stationary periods and begins withdrawing southward in September (Ding, 2004; Hsu, 2005). The rainy seasons of EASM, including the pre-summer rainy season over South China, mei-yu (in China) normally occurs during the stationary periods, which are imbedded in the northward advance of the summer monsoon. The anomaly of EASM could cause floods and droughts which are crucial in the livelihood of more than one billion people (Gu et al., 2015; Webster et al., 1998; Yu et al., 2018). However, the manner in which climatological rainfall and interannual variation of EASM can be reliably reproduced remains a challenge because of the complex topography and model limitation. Coupled model intercomparison project phase 3 (CMIP3) and CMIP5 have problems simulating precipitation in this region. Recent studies have suggested that the new generation of GCMs from CMIP5 archive exhibits several improvements to reproduce the climatology and interannual variability of EASM compared with the CMIP3 GCMs, although the simulated biases remained and large intermodel spread existed (Chen and Bordoni, 2014; Gu et al., 2015; Huang et al., 2013; Yang et al., 2017). For example, the mei-yu rainfall band in GCMs is missing even though the monsoon circulation is well reproduced.

Considering these deficiencies, high-resolution GCMs have been developed to improve the capabilities in monsoon features simulation, including orographic precipitation, low-level jet orientation and variability, as well as the mei-yu onset and withdrawal (Kitoh et al., 2013; Kusunoki et al., 2006). However, these experiments are remained burdensome due to large computational cost required for multi-decadal simulations. Therefore, the regional climate models (RCMs) focusing on a region of interest are commonly used in regional climate projection and climate change impacts studies (Gao et al., 2006; Giorgi and Mearns, 1999; Gu et al., 2012; Wang et al., 2004; Yira et al., 2017; Yu et al., 2006). The resolution of RCMs is approximately 12-50 km, and it could consider local scale forcing, e.g. complex terrain features and land cover heterogeneities in a physically based method. However, RCMs inherit the biases from systematic model errors because of the imperfect conceptualization, discretization, and spatial averaging within grid cells (Dong et al., 2018). Nonetheless, RCM ensembles can be used to understand and characterize uncertainties from different sources, such as future climate scenario, the driving GCM and regional model physics, and therefore, reduce the uncertainties and increase credibility in future projections. The ongoing coordinated regional downscaling experiment (CORDEX) aims to provide high-resolution future regional climate projections for the majority of populated land regions globally by using multi-RCMs and to present an interface for applicants of climate simulations in climate change impact, adaptation, and mitigation studies (Giorgi et al., 2009; Jones et al., 2011). The CORDEX in East Asia ( CORDEX-EA) is the East Asian branch of the CORDEX experiment, and it provides ensemble regional climate

simulations (https://cordex-ea.climate.go.kr/main/modelsPage.do). A series of studies based on RCMs within CORDEX-EA has been conducted to project extreme and mean precipitation and temperature in East Asia (Jin et al., 2016; Lee et al., 2014; Niu et al., 2015; Park et al., 2016; Tang et al., 2016; Um et al., 2017), but little attention has been paid to quantify the contributions of the uncertainty in future climate projection over China.

Despite large improvements in the simulation of local processes, future climate projections are still accompanied by large uncertainties stemming from different sources, including the forcing GCMs, emission scenarios, downscaling methods (RCMs or statistical downscaling methods), and natural climate internal variability (Déqué et al., 2007; Deser et al., 2012). Numerous studies have demonstrated that GCMs are the main source of uncertainty (Seo et al., 2016). Other uncertainty sources, such as RCMs and internal variability will become more important than GCMs after excluding the outliers from the GCM ensemble

(Kay et al., 2009; Wilby and Harris, 2006). In a non-stationary climate, the internal variability of a given GCM-RCM chain can remain high above the trend related to a given emission scenarios forcing (Lafaysse et al., 2014; O'Brien et al., 2011). Little attention has been devoted to quantify the contributions of the uncertainty arising from RCMs and internal variability in future climate projection over China. Objectively evaluating the capability of RCMs and quantifying uncertainty in future climate projections are necessary.

In this study, we evaluate the performance of five RCMs within CORDEX-EA to reproduce present-day climate and analyse the projected future climate changes under the middle emission scenario. More importantly, biases in current climate simulations and uncertainties in future climate projections attributed to RCMs and internal variability are further analysed. This paper is structured as follows. Data from observation and model simulation, and analysis method are described in the succeeding section. Section 3 presents the historical performances of RCMs for temperature and precipitation and future

climate changes under RCP4.5 emission scenario in China. The uncertainties in regional future climate projection caused by inter-RCMs and natural climate internal variability are also discussed. The summary and conclusion are presented in Section 4.

## 2 Data and methods

### 2.1 Observations

The reference temperature data used to evaluate the model results with observation data develops from the Climate Research Unit Timeseries 3.23 (CRU) of the University of East Anglia, with a spatial resolution of 0.5°, derived from gauge measurements (Harris et al., 2014). Meanwhile, the reference precipitation data, namely the Asian Precipitation-Highly Resolved Observational Data Integration Toward Evaluation (APHRODITE, hereafter APHRO) dataset with a spatial resolution of 0.25° was used to evaluate RCMs (Yatagai et al., 2012). To facilitate the comparison, outputs from a host of

RCMs were converted to a common grid of $0.5° \times 0.5°$ latitude/longitude as remapped to the CRU and APHRO observations, using bilinear interpolation. The reasons why CRU and APHRO products are used as reference in this study are clarified as below.

Some studies have focused on comparing and evaluating the spatial-temporal similarities and differences of several widely used observed gridded datasets over China (Sun et al., 2014; Wu and Gao, 2013; Yin et al., 2015). Among the widely used gridded dataset, such as the Global Precipitation Climatology Centre (CPCC) product, the University of Delaware (UDEL) product, CRU data and the National Meteorological Information Center dataset from China Meteorological Administration, all temperature datasets exhibit similar distribution patterns for the annual average temperature in mainland China. Considering its easier access and wider usage in the evaluation of RCMs used in China and East Asian (Wang et al., 2017), CRU product is used as the reference temperature data in this study. APHRO's daily gridded precipitation, presently the only long-term, continental-scale, high-resolution daily product, is constructed based on the data collected at 5000-12000 stations, which represent 2.3-4.5 times the data made available through the stations used for generating global gridded (i.e. CRU, GPCC and UDEL) (Yatagai et al., 2012). Thus, the APHRO dataset would give more confidence in the robustness of the results in comparison with other global precipitation datasets and thus is widely used for evaluating the performance of RCM in East Asia (Gao et al., 2017; Lau et al., 2017; Um et al., 2017). In addition, the CRU and APHRO product are used instead of station data accessible from China Meteorological Administration, owing to the study area involving in the domain of East Asia extending beyond China territory.

## 2.2 Models and experiments

In this study, we used five RCMs, namely, HadGEM3-RA, MM5, WRF, RegCM4, and RSM for East Asian climate experiments (Table 1). They are derived from the CORDEX East Asia experiment that is able to provide a global holistic framework for regional climate projections so as to understand their uncertainties as well as provide model evaluation. Moreover, the selected five RCMs have been demonstrated to have abilities to reproduce the regional climate over East Asia and have been used for modelling and predicting extreme climate as well as investigating physical processes of East Asia climate (Cha and Lee, 2009; Cha et al., 2011; Hong and Yhang, 2010; Park et al., 2008; Yhang and Hong, 2008). The spatial resolution of the data is 50 km (except HadGEM3-RA is 0.44°), and the whole CORDEX-EA domain includes East Asia, India, the Western Pacific Ocean, and the northern part of Australia, as shown in Figure 1. Model configurations including physical schemes are summarized in Table 1. Please refer to the references Suh et al. (2012) and Park et al. (2016) about more details about RCMs used in this study.

**Table 1**

**Figure 1**

In this study, two types of current climate experiments from five RCMs were performed, including the evaluation (hereafter EVAL) experiment from 1989 to 2008 and the historical (HIST) experiment from 1980 to 2005. The EVAL experiment acquires initial and boundary conditions from the National Centers for Environmental Prediction reanalysis, whereas the HIST experiment is forced by the Atmosphere-Ocean coupled Hadley Center Global Environmental Model version 2 (HadGEM2-AO) simulation. HadGEM2-AO (1.875°×1.25° horizontal resolution) has been used for climate

simulations in a CMIP5 set of long-term experiments and has been demonstrated to have a reasonable ability to capture the East Asian climatology (Baek et al., 2013; Martin et al., 2011; Sperber et al., 2013). Future climate simulation is driven from the HadGEM2-AO under RCP 4.5 scenario, which is an intermediate scenario and a cost-minimizing pathway that total radiative forcing is stabilized at 4.5 W m$^{-2}$ in the year 2100 (Thomson et al., 2011). The reference period from 1980 to 1999 and the scenario period from 2030 to 2049 are analysed for climate changes research in this study.

The multi-model ensemble (MME) mean, defined as the pointwise arithmetic average over all individual model climatologies, narrows down inter-RCM uncertainties because of their differences in model structures and physics. To further evaluate the model performance on smaller spatial scales, we evaluate the performance of RCMs over five selected sub-regions (as shown in Figure 1), namely, Northeast China (40-50°N, 115-130°E), North China (30-40 °N, 105-120 °E), South China (22-30 °N, 105-120 °E), Northwest China (35-45 °N, 80-95 °E), and Tibetan Plateau (28-35 °N, 80-95 °E).

## 2.3 Analysis methods

The root-mean-square error (RMSE), bias and Taylor diagram analysis are selected for statistical measurements of the performance for the individual RCM and the MME. The former two indexes are used for evaluating the ability of models in reproducing annual and seasonal mean of climatology. The Taylor diagram is designed to quantify the degree of correspondence between the modelled and observed behavior by plotting a 2D graph with three statistics (correlation coefficient, standard deviation, and RMSE). In the Taylor diagram, a small distance between reference and compared objects indicates close agreement (Baker and Taylor, 2016; Sun et al., 2015). In general, the Taylor diagram enable statistics for different fields (with different units) to show in a single plot, facilitating the comparative assessment of different models (Taylor, 2001).

Uncertainty in projected climate change mainly arises from the internal variability of the climate system, the model uncertainty, and the scenario uncertainty (Niu et al., 2015; Woldemeskel et al., 2016). In this study, all RCMs are driven by the same GCM under the same scenario, and thus, the uncertainty of the climate projections is mainly caused by inter-RCM and internal variability. The method developed by Hawkins and Sutton (2009; 2011) is used to separate these two sources of uncertainty. Here we give a brief illustration.

(1) Firstly, a smooth fourth-order polynomial is used to fit each individual simulation over the years 1980-2049 by using ordinary least squares method. Then the raw simulation of each model $X_{m,t}$ for the model $m$ and year $t$ can be expressed by

$$X_{m,t} = z_{m,t} + c_m + \varepsilon_{m,t} \tag{1}$$

where $z_{m,t}$ represents the simulation from the smooth fit for the model $m$ and year $t$ minus the reference data; the reference data is denoted by $c_m$, and the residual (internal variability) is denoted by $\varepsilon_{m,t}$. Here the reference data is the mean of simulation from the smooth fit during the years 1980-1999.

(2) The RCMs are weighted by their performance in simulating the current climate from the mean of 1980-1999, up to year 1999. Thus, each model is weighted according to

$$w_m = \frac{1}{z_{obs} + |z_{m,1999} - z_{obs}|} \tag{2}$$

where $z_{m,1999}$ is the model climate changes in 1999 relative to 1980-1999, and $z_{obs}$ is an observational estimate derived from fitting a similar fourth-order polynomial to observations. The normalized quantities ($W_m$) of these weightings can be expressed as

$$W_m = \frac{w_m}{\sum_m w_m} \tag{3}$$

(3) Internal variability ($V$, as shown in Eq. 4) is defined as the multi-model mean of the variance of the residuals from the fit for each model,

$$V = \sum_m W_m \, \mathrm{var}_t(\varepsilon_{m,t}) \tag{4}$$

$$M(t) = \mathrm{var}_m^w(z_{m,t}) \tag{5}$$

(4) Intermodel variability ($M$, as shown in Eq. 5) is estimated from the weighted variance ($\mathrm{var}^w$) in different RCM prediction fits ($z_{m,t}$), where $\mathrm{var}_t(.)$ and $\mathrm{var}_m(.)$ indicate the variance across time and model, respectively.

(5) It was assumed that the two sources of uncertainty can be treated independently (i.e., no interaction exists between them). Thus, the total variability $V_T$ is

$$V_T(t) = V + M(t) \tag{6}$$

(6) The fraction of variance of internal variability and model uncertainty defined as $V/V_T(t)$ and $M(t)/V_T(t)$, respectively.

## 3 Results

### 3.1 Climatology for the historical climate

#### 3.1.1 Historical annual average climate evaluation

Figure 2 shows the annual average temperature of CRU, the driving GCM HadGEM2-AO and multi-model ensemble, as well as the temperature biases of five RCMs driven by HadGEM2-AO from 1980 to 2005. Obviously, both the MME and five RCMs can capture the spatial pattern of annual mean temperature in China, with a decreasing south-north gradient and a cold area in the Tibetan Plateau. Moreover, the MME presents overall best results to reproduce the temperature spatial distribution and provides less than 1 °C temperature biases over most area in China. However, all RCMs generally overestimated the mean temperature over most of the domain, in particular warmer mean temperature is simulated by MM5 and HadGEM3-RA. The only exception is that RSM underestimated the mean temperature over the Tibetan Plateau.

**Figure 2**

The RCMs provide reasonably accurate simulations for mean temperature during the historical period, but they are less successful at reproducing precipitation. Figure 3 shows the annual average precipitation from APHRO, HadGEM2-AO and

MME, as well as the precipitation biases from five RCMs in the current period. It is found that the spatial pattern for annual mean precipitation is characterised by a decreasing southeast-northwest gradient over China, which can be successfully simulated by all RCMs. However, quite large precipitation biases are found in different RCMs. For instance, WRF underestimated the annual mean precipitation in northwest China, where mean precipitation was overestimate by the other RCMs. In comparison with the simulation from each RCM, the MME is better in reproducing annual mean precipitation over most subregions in China.

**Figure 3**

The spatial variability statistics of the models in reproducing the annual mean temperature and precipitation by the Taylor plot (Taylor, 2001) are exhibited in Figure 4. The temperature simulations of the five RCMs exhibit a good spatial pattern correlation, ranging from 0.83 to 0.96, whereas the precipitation simulation show a relatively extensive range of spatial pattern correlations from 0.29 to 0.93. Besides, the MME is superior to most RCMs in capturing spatial variability of these climate variables, as reflected by higher spatial correlation coefficient and lower RMSE. Several reasons could explain this phenomenon, as also noted by other scholars in their studies on model inter-comparisons (Huttunen et al., 2017; Phillips and Gleckler, 2006; Rozante et al., 2014). On the one hand, the bias of a simulated climate field is symptomatic of random errors to a certain extent, and the MME may reduce or counteract this error from the RCM. On the other hand, the pointwise variations of the climate field are smoothened out by averaging, thereby filtering regional-scale simulations, which current climate models are difficult to capture.

**Figure 4**

### 3.1.2 Interannual and seasonal variability

The ability of a climate model to capture realistic interannual variability is critical measure of its performance. The time series of the annual mean temperature and precipitation from RCMs are compared with CRU and APHRO in Figure 5. Evidently, the interannual variation of the climatology is generally well reproduced in the MME. In the evaluation experiment for 1989-2008, the correlation coefficient of the annual climatology time series at five subregions between the observation and simulation from the MME ranges from 0.52 to 0.78 for temperature and from 0.50 to 0.87 for precipitation. The correlation coefficient is always lower in West China compared with that in East China, especially in Tibetan Plateau. In the historical experiment from 1980 to 2005, the MME show better performance, in comparison with the RCMs which have difficulty in reproducing the interannual variability for precipitation because of the impact of the driving GCM.

**Figure 5**

The temporal distributions of precipitation and temperature throughout the year are important for ecosystems and water resource management. To evaluate the RCM's ability to capture the seasonal variability of climatologies, the seasonal cycles of simulated temperature and precipitation averaged over five subregions in China are examined (Figure 6). It is evident that the seasonal pattern of precipitation is featured by one peak in June over south China and in July over the rest regions, which can be successfully reproduced by all RCMs and MME. However, the inter-model difference in simulated precipitation is

large. For instance, monthly precipitation is always underestimated by WRF and overestimated by MM5 and HadGEM3-RA, especially larger bias is shown in summer. Among five RCMs, RegCM is the one with best ability to simulate the seasonal cycles of precipitation. The MME generally provide the most accurate simulation for the temporal distribution of precipitation, in comparison with the RCMs. As for the temperature, the RCMs can capture its temporal pattern over all subregions. Moreover, mean temperature in different months are always overestimated by most RCMs. However, the MME reduces the bias from the RCMs and therefore generate more accurate temporal distribution for mean temperature.

**Figure 6**

### 3.1.3 The added value for RCMs

The added values for high-resolution RCMs were confirmed by comparing the performance of RCMs and driving GCM HadGEM2-AO in reproducing annual mean precipitation and temperature during the historical period. According to the Figure 4-6, it is found that the added value for RCMs depends largely on the climate variable and the area of interest. The added value of the RCMs in comparison with the driving GCM was evident in term of annual mean temperature over all five subregions, with higher spatial and temporal correlation coefficient and less seasonal bias for all five RCMs. Compared with the driving GCM simulations, the historical precipitation over South China, Northwest China and the Tibetan Plateau were improved in most RCMs. The exceptions are over Northeast China and North China where higher performance is shown for the driving global climate model. In reality, the added value in RCM simulations is mainly concerned with a better representation of spatial variability of surface climate statistics, particularly in areas with small-scale land surface forcing such as orographic and coastal features. Thus, the added value in RCM simulations is commonly significant in regions with fine-scale surface forcing, whereas the performance of RCM is less improved or even worse than that of the driving GCM over relatively flat regions. For instance, Prommel and Geyer (Prömmel et al., 2010) also found the RCM deteriorates some results compared to the driving GCM in relatively flat regions surrounding the Alps, especially in summer. In most cases, five RCMs perform better than the driving GCM HadGEM2-AO. It needs to be emphasized that the better model performance tends to increase confidence in the future climate projections from RCMs.

## 3.2 Multi-RCM future climate projection

### 3.2.1 Future change in climatology

According to figure 7 showing the projections for mean temperature from the driving GCM, RCMs and the MME, similar warming trends are detected over the entire domain from 2030 to 2049 under RCP4.5 scenario. All five models project substantially significant warming while exhibiting different spatial patterns. The increases in annual temperature by the MME are 1.3, 1.0, 0.9, 1.2, and 1.3 °C over the Northeast, North, South, Northwest, and Tibetan Plateau subregions, respectively. The warming in northern and western China is more significant than that in southern China, especially in Northeast China and Tibetan Plateau, which is similar to the results from previous studies (Huttunen et al., 2017; Phillips and Gleckler, 2006;

Rozante et al., 2014). Moreover, the magnitude for the increase in annual temperature over a given subregion varies with the RCM. For instance, the projected increase in mean temperature over the Tibetan Plateau ranges from 0.9 °C to 1.6 °C.

Figure 7

Figure 8 shows the spatial distributions of changes in annual mean precipitation (RCP4.5-baseline). During the period 2030-2049, increased precipitation is projected by the MME and most RCMs over China. Moreover, the projected spatial pattern from the driving GCM, the MME and RCMs is nearly consistent, with the most prominent increase in precipitation over the north and northwest China and slightly increase precipitation over the rest regions. The only exception is the results from WRF, by which the declined mean precipitation is projected over China. In particular, wider range for the change in projected annual precipitation are shown over the Tibetan Plateau. This is related to the fact that significant difference in projected precipitation change between WRF and the other RCMs. Therefore, the projected change in annual precipitation over the Tibetan Plateau should be treated with caution. Besides, opposite signals for projected changes in average precipitation between the MME and the driving GCM are detected over South China, Northeast China and Tibetan Plateau (Table 2). Particularly the differences in projection form two methods above are largest at the Tibetan Plateau, up to about 10%.

Figure 8

Table 2

### 3.2.2 Change in seasonal cycle

The future changes of temperature and precipitation are characteristic of regionality and seasonality. The ensemble projection (Figure 9) indicates that the monthly temperature change over five subregions in China ranges from 0.3 °C to 2.2 °C under the RCP4.5 scenario. A more remarkable warming in cold months from November to March is detected by all RCMs. The seasonal cycle of temperature change in MME is also similar to that of the driving GCM HadGEM2-AO. Most RCMs project positive monthly precipitations changes for summer (from June to August) over China, with the exception of the Tibetan Plateau. However, the projected monthly precipitation change by MME has larger magnitude and variability than the driving GCM. This phenomenon concerns the significance of the model physics and processes for future climate projection. The configurations of each RCM were showed in Table 1. For each RCM, optimal schemes of the dynamical and physical processes were determined through the model sensitivity analysis (Suh et al., 2012). In general, convective parameterization is one of the most important and sensitive process in a RCM (Huang and Gao, 2017). Land surface parameterization, as well as those parameterizations over the ocean, are also very important because they control the quantity of water vapor flux entering into atmosphere from the earth's surface (Zhao and Li, 2015). Thus, the phenomenon above could be attributed to the difference in convective parameterization, land surface parameterizations, as well as those parameterizations over the ocean between GCMs and RCMs. On the other hand, the discrepancies between the RCMs and the driving GCM indicate that the RCM projections are sensitive to local and regional processes and the corresponding methods incorporated in the model (Diallo et al., 2012; Saini et al., 2015).

Figure 9

### 3.2.3 Inter-RCM variability of multi-RCM projections

The uncertainties of regional climate projection arise from the GCMs, emission scenarios, RCMs, and internal variability for natural climate. In this study, the regional future climate is projected by using five RCMs forced with the same GCM under an intermediate scenario (RCP4.5). Consequently, the contribution of inter-RCM variability and internal variability to total uncertainty in the projections are analysed in this section.

The contributions of the model uncertainty and natural climate internal variability to the total prediction uncertainty are estimated by the method proposed by Hawkins and Sutton (2009). The results for five subregions are shown in Figure 10. The relative importance of model uncertainty increases with prediction lead time over all subregions. For temperature, the model uncertainty is the primary source of uncertainty over the northeast, northwest China, and Tibetan Plateau from 2030 to 2049, reaching up to 70%. The model uncertainty minimally contributes (approximately 40%) to the total uncertainty over north and south China before the middle of the 21$^{st}$ century. For the uncertainty on projected precipitation, the internal variability is the dominant uncertainty source over most regions, except for the Tibetan Plateau where the model uncertainties reached up to 60%. The uncertainties come from the driving GCM, and the emission scenarios are not discussed in this study although they have been recognized as important components for total uncertainty (Déqué et al., 2012). Further research on uncertainty quantification on the basis of different GCMs, RCMs and emission scenarios are needed in the future.

**Figure 10**

### 4 Summary and conclusions

In this research, five RCM models, which are simulated within the CORDEX-EA initiative at 50km resolution with boundary forcing from a CMIP5 global model applying the RCP4.5 scenario, are employed to derive the future climate change signal for China and five selected smaller investigation areas. Meanwhile, the contribution of the model uncertainty and natural climate internal variability to the total prediction uncertainty are quantified.

The control runs of CORDEX-EA RCMs revealed an overall reasonable representation of the mean climate properties when compared with the observational gridded dataset. All RCMs generally provide warm biases, whereas the MME demonstrates the overall best performance, with less than 1 °C annual average temperature biases over most area in China. Similarly, the MME outperformed the individual RCM in reproducing the observed spatial pattern of precipitation. Moreover, five RCMs perform better than the driving GCM HadGEM2-AO in reproducing annual and seasonal precipitation over most subregions. Therefore, it is concluded that the MME constructed based on the set of RCMs from CORDEX-EA can be used to provide useful information on climate projections over East Asia.

For 2030 to 2049, MME indicated consistent warming ranging from 0.9 °C to 1.6 °C in the entire domain and more pronounced warming was detected in northern and western China. Seasonal temperature changes drastically in cold months, which is similar to that of the driving GCM. Besides, the annual precipitation is likely to increase in most of subregions. The projected spatial pattern for annual precipitation is characterized by prominent increase over the north and northwest China

and slightly increased precipitation over the rest regions. Moreover, precipitation in summer months are predicted to consistently increase over the entire domain, with the exception of the Tibetan Plateau. It should be noted that the projected monthly precipitation change by MME has larger magnitude and variability than the driving GCM.

This study identified the contributions of model uncertainty and internal variability. The uncertainty in projected temperature mainly arises from the internal variability over north and south China. Whereas, the model uncertainty is clearly dominant over the rest three subregions, explaining approximately 70% of the total uncertainty. For precipitation, the internal variability is dominant over most regions except for the Tibetan Plateau, in which model uncertainties reach up to 60%. Model uncertainty also increases with prediction lead time over all subregions. RCM simulation results are also influenced by the internal physics and boundary conditions from GCMs as discussed in other's studies (Mariotti et al., 2011; Syed et al., 2012). More reliable future climate information and uncertainty quantification could be provided by coupling large ensemble of GCMs and RCMs under different emission scenarios.

**Acknowledgments**

This work was supported by the National Key R&D Program of China (Grant No. 2016YFC0402706, 2016YFC0402710), the National Natural Science Foundation of China (No. 41501015, 51539003, 51421006, 51509263), the Fundamental Research Funds for the Central Universities (2016B00114). We acknowledge the CORDEX-East Asia Databank, which is responsible for the CORDEX dataset, and we thank the National Institute of Meteorological Research (NIMR), three universities in the Republic of Korea (Seoul National Univ., Yonsei Univ., Kongju National Univ.) and other cooperative research institutes in East Asia region for producing and making available their model output. We also thank two anonymous reviewers for their constructive and insightful comments that helped to improve the original manuscript.

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

**Table Captions**

**Table 1**. RCMs used in this study.

**Table 2**. The future changes in average temperature (T; °C) and precipitation (P; %) for the five subregions. The ensemble averages for each statistic are given in the second line. The projections by the forcing GCM are given in the last line.

**Table 1. RCMs used in this study[a] (Park et al., 2016)**

|  | HadGEM3-RA | RegCM4 | MM5 | WRF | RSM |
|---|---|---|---|---|---|
| Resolution | 0.44° | 50km | 50km | 50km | 50km |
| Dynamic process | Non-hydrostatic | Hydrostatic | Non-hydrostatic | Non-hydrostatic | Hydrostatic |
| Convective scheme | Revised mass flux scheme | MIT-Emanuel | Kain-Fritch II | Kain-Fritch II | Simplified Arakawa-Schubert |
| Land surface parameterization | MOSES2 | CLM3 | CLM3 | NOAH | NOAH |
| Planetary boundary layer | MOSES2 non-local | Holtslag | YSU | YSU | YSU |
| Spectral nudging | No | Yes | Yes | Yes | Yes |
| Center of research | MOHC | ICTP | NCAR | NCAR | YSU |
| References | Davies et al.(2005) | Giorgi et al.(2012) | Cha and Lee(2009) | Skamarock et al.(2005) | Hong et al.(2013) |

[a]MOSES= Met Office Surface Exchange Scheme, CLM= Community Land Model, NOAH=Noah Land Surface Model, YSU= Yonsei University scheme, MOHC= The Met Office Hadley Centre, ICTP= The International Centre for Theoretical Physics, NCAR= National Center for Atmospheric Research

**Table 2. The future changes in average temperature (T; °C) and precipitation (P; %) for the five subregions. The ensemble averages for each statistic are given in the second line. The projections by the forcing GCM are given in the last line.**

|  |  | WRF | MM5 | HadGEM3-RA | RegCM | RSM | Ensemble | HadGEM2-AO |
|---|---|---|---|---|---|---|---|---|
| Northeast China | T(°C) | 0.2 | 2.7 | 1.4 | 1.4 | 1.1 | 1.3 | 0.8 |
| | P(%) | -21.7 | 8.2 | 13.0 | 4.4 | 7.1 | 1.5 | -0.4 |
| North China | T(°C) | 0.3 | 1.7 | 1.1 | 1.0 | 1.0 | 1.0 | 0.8 |
| | P(%) | -1.5 | 15.1 | 3.1 | 10.2 | 3.3 | 6.1 | 4.9 |
| South China | T(°C) | 0.5 | 1.5 | 1.0 | 0.8 | 0.8 | 0.9 | 0.7 |
| | P(%) | -14.6 | -1.6 | 4.8 | 4.9 | 1.3 | -1.5 | 2.3 |
| Northwest China | T(°C) | 1.3 | 0.8 | 1.5 | 1.3 | 1.1 | 1.2 | 1.2 |
| | P(%) | -27.0 | 19.4 | 2.2 | 4.7 | 8.9 | 3.6 | 7.2 |
| Tibetan Plateau | T(°C) | 0.9 | 1.4 | 1.2 | 1.3 | 1.6 | 1.3 | 1.4 |
| | P(%) | -31.6 | -17.8 | 2.4 | 6.4 | 7.4 | -7.8 | 2.1 |

**Figure Captions**

**Figure 1**. The simulation domain of CORDEX-EA and the topography of the regional climate models (m). The boxes illustrate the five selected subregions over China: Northeast China (NE), North China (NC), South China (SC), Northwest China (NW), and Tibetan Plateau (TP).

**Figure 2**. Spatial distributions of annual average temperature (ºC) from CRU (a), the driving GCM HadGEM2-AO (b), multi-model ensemble (c), and temperature biases (ºC) of the driving GCM HadGEM2-AO (d), multi-RCM ensemble (e, f) and five RCMs (g-k) during 1980-2005.

**Figure 3**. Spatial distributions of annual average precipitation (mm/year) from APHRO (a), the driving GCM HadGEM2-AO (b), MME (c), and precipitation biases (%) of the driving GCM HadGEM2-AO (d), MME (e) and five RCMs (f-j) during 1980-2005.

**Figure 4**. The Taylor diagram to evaluate the skill of the models in reproducing the annual average temperature and precipitation over the five regions of China, using the CRU (for temperature) and APHRO (for precipitation) data as the reference. The azimuthal axis shows the pattern spatial correlation. The redial distance from the origin represents the spatial variability, whereas the distance from the OBS point is the centred RMSE difference between the simulated and observed.

**Figure 5**. Temporal evolution of the annual mean temperature (left two panels) and precipitation (right two panels) in RCM simulations and observation over the five subregions during the 1989-2008 (EVAL) and 1980-2005 (HIST) periods. The correlation coefficient between RCMs ensemble and the observation are shown at the top right of each panel.

**Figure 6**. Observed and simulated monthly mean temperature and precipitation over the five subregions during the 1989-2008 (EVAL) and 1980-2005 (HIST) periods.

**Figure 7**. Projected future changes (RCP4.5-Baseline) in surface air temperature by the forcing GCM HadGEM2-AO, the MME and each of the five RCMs.

**Figure 8**. Projected future changes ((RCP4.5-Baseline)/Baseline×100%) in precipitation by the forcing GCM HadGEM2-AO, the MME and each of the five RCMs.

**Figure 9**. Projected future changes in monthly mean temperature and precipitation by the forcing GCM HadGEM2-AO, the MME and each of the five RCMs under RCP4.5 scenario.

**Figure 10**. Fraction of total variance in future temperature (left panel) and precipitation (right panel) projections explained by intermodel variability (gray) and internal variability (white) over the five subregions.

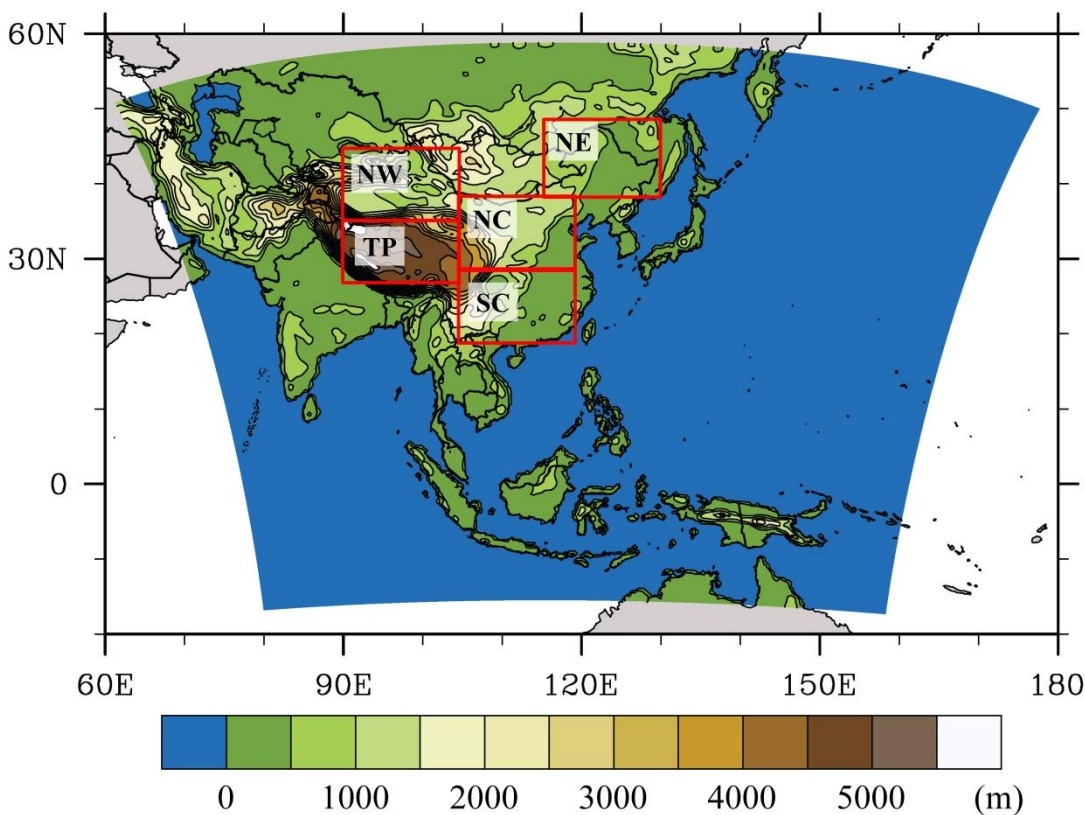

**Figure 1.** The simulation domain of CORDEX-EA and the topography of the regional climate models (m). The boxes illustrate the five selected subregions over China: Northeast China (NE), North China (NC), South China (SC), Northwest China (NW), and Tibetan Plateau (TP).

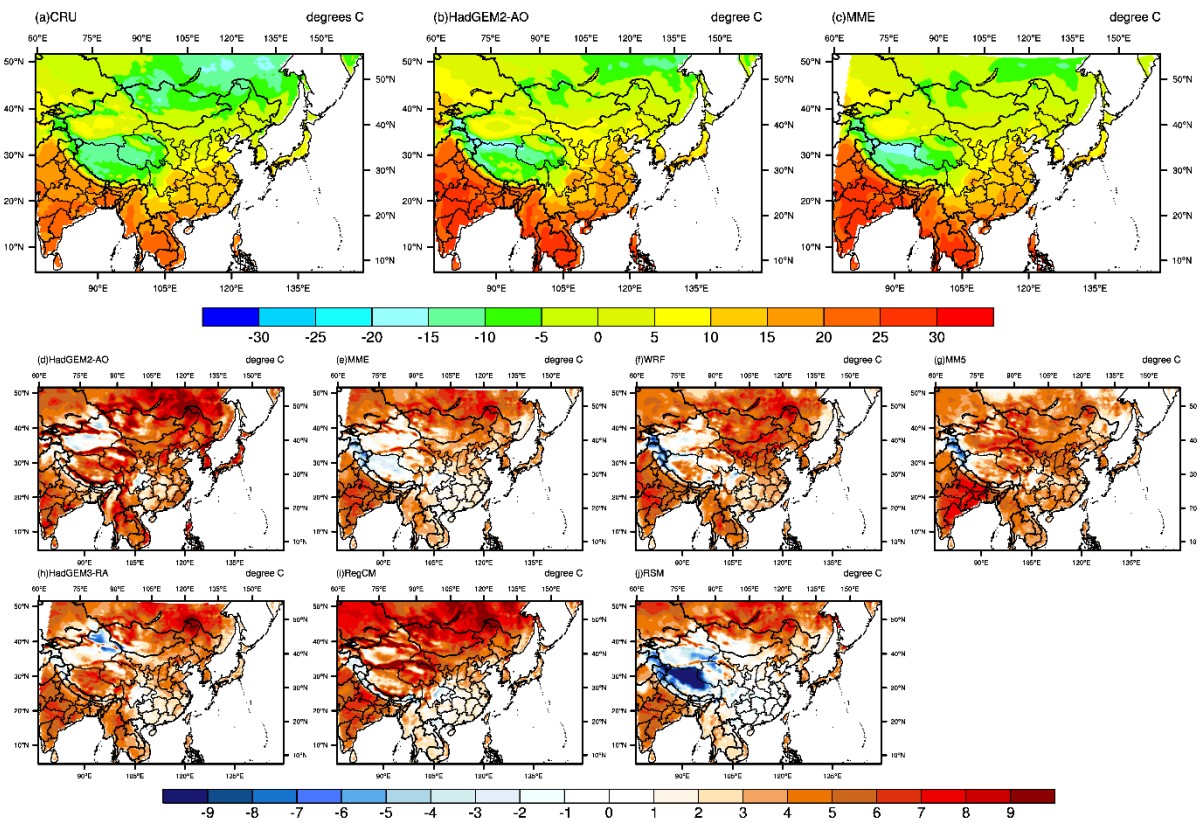

**Figure 2.** Spatial distributions of annual average temperature (ºC) from CRU (a), the driving GCM HadGEM2-AO (b), multi-model ensemble (c), and temperature biases (ºC) of the driving GCM HadGEM2-AO (d), multi-RCM ensemble (e, f) and five RCMs (g-k) during 1980-2005.

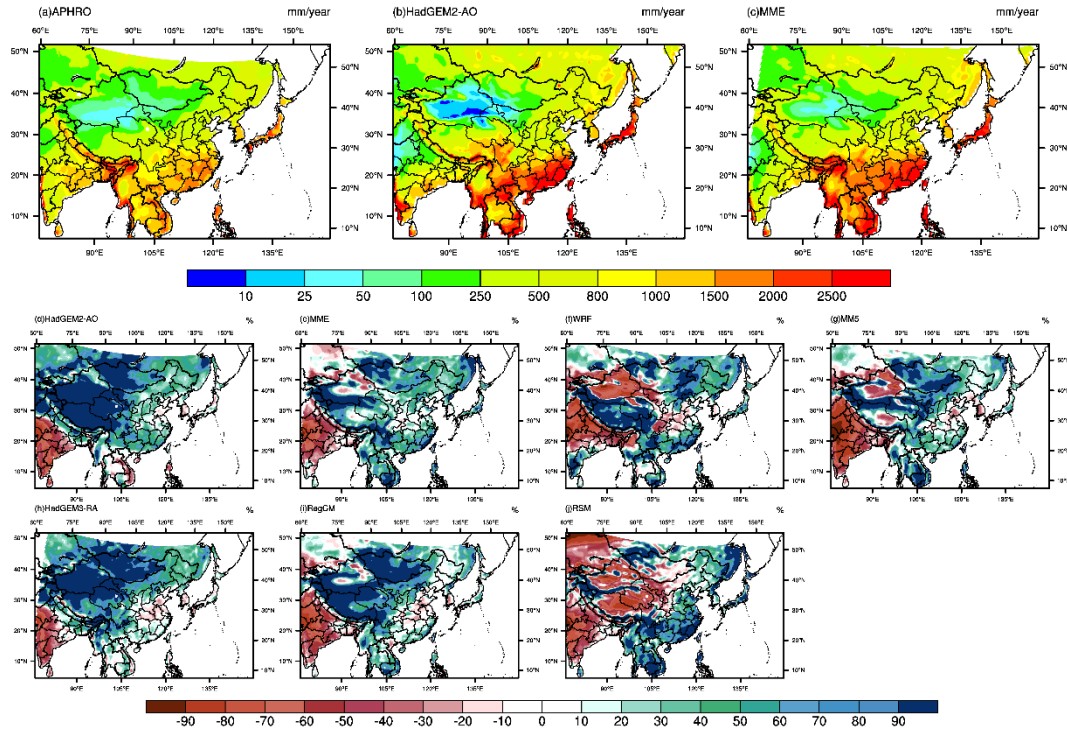

**Figure 3.** Spatial distributions of annual average precipitation (mm/year) from APHRO (a), the driving GCM HadGEM2-AO (b), MME (c), and precipitation biases (%) of the driving GCM HadGEM2-AO (d), MME (e) and five RCMs (f-j) during 1980-2005.

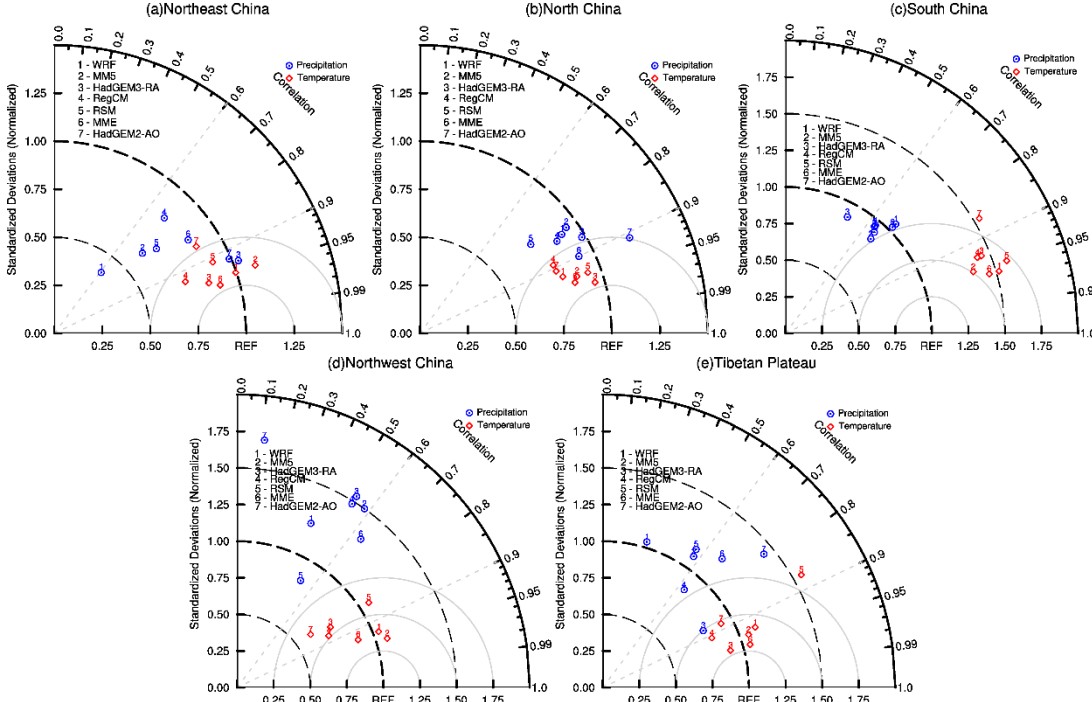

**Figure 4.** The Taylor diagram to evaluate the skill of the models in reproducing the annual average temperature and precipitation over the five regions of China, using the CRU (for temperature) and APHRO (for precipitation) data as the reference. The azimuthal axis shows the pattern spatial correlation. The redial distance from the origin represents the spatial variability, whereas the distance from the OBS point is the centred RMSE difference between the simulated and observed.

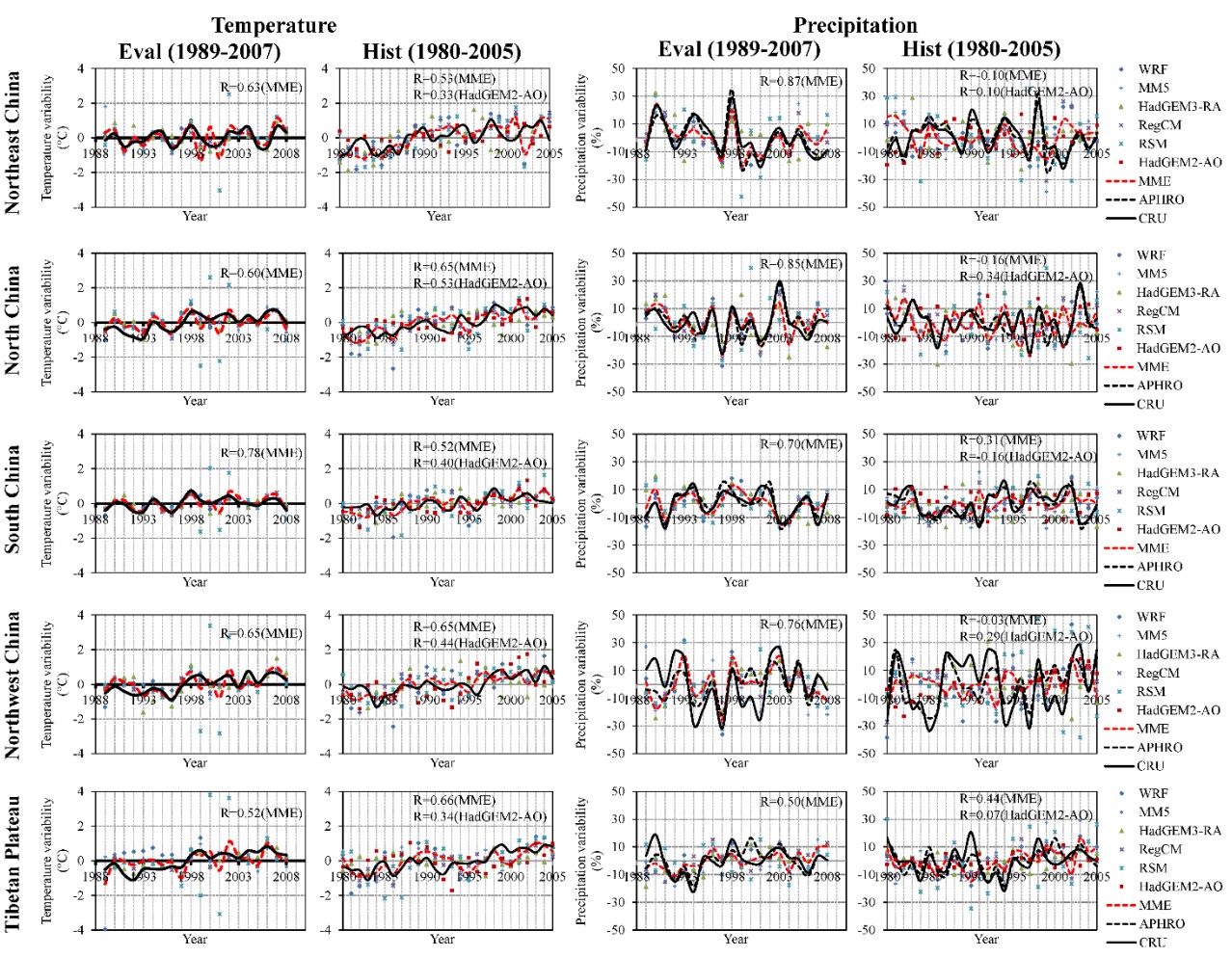

**Figure 5.** Temporal evolution of the annual mean temperature (left two panels) and precipitation (right two panels) in RCM simulations and observation over the five subregions during the 1989-2008 (EVAL) and 1980-2005 (HIST) periods. The correlation coefficient between RCMs ensemble and the observation are shown at the top right of each panel.

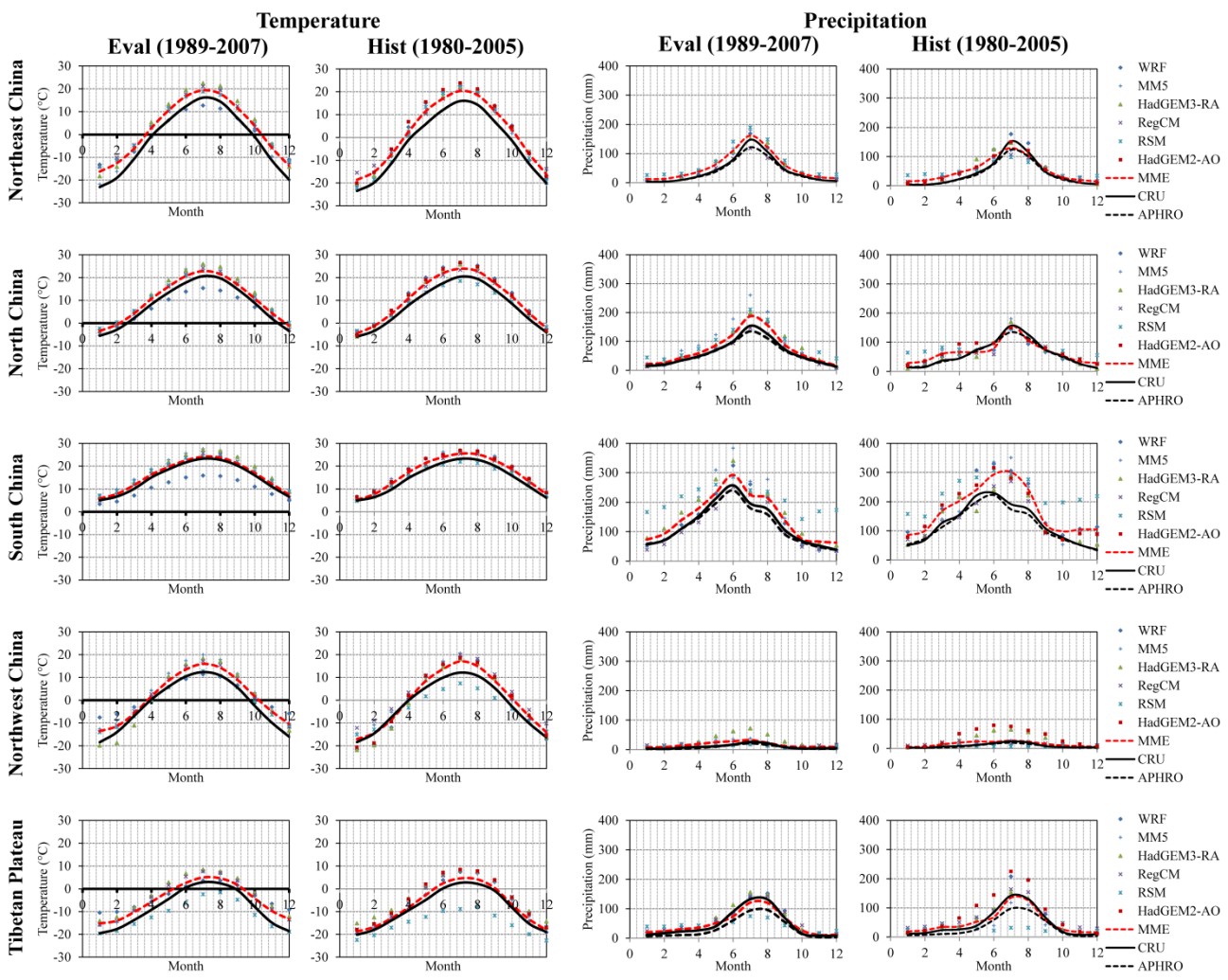

**Figure 6.** Observed and simulated monthly mean temperature and precipitation over the five subregions during the 1989-2008 (EVAL) and 1980-2005 (HIST) periods.

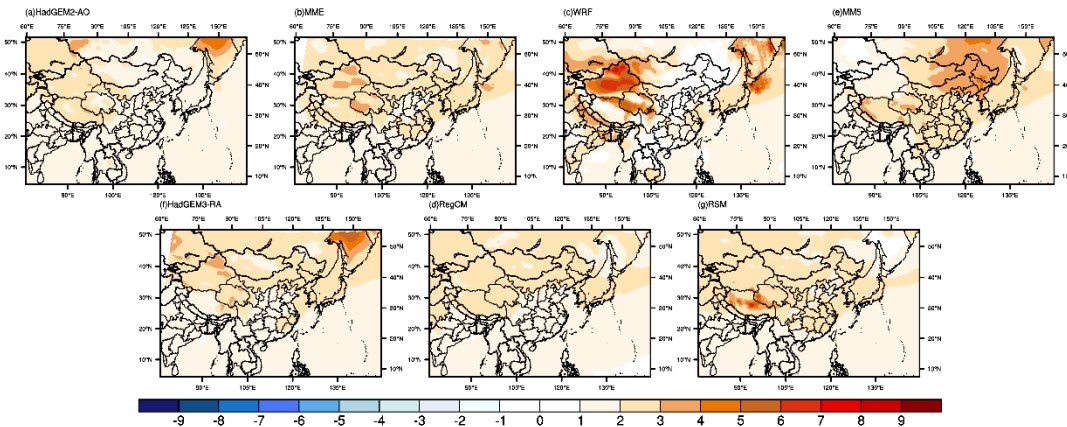

**Figure 7.** Projected future changes (RCP4.5-Baseline) in surface air temperature by the forcing GCM HadGEM2-AO, the MME and each of the five RCMs.

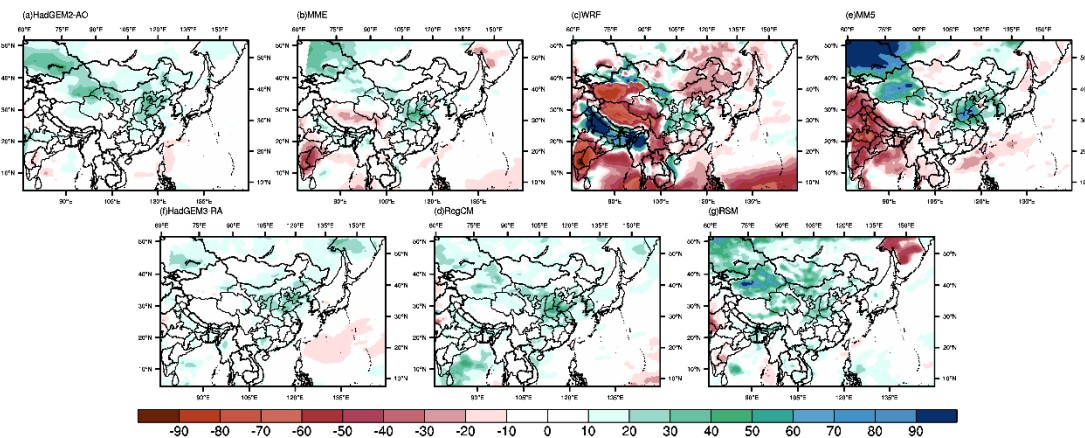

5 **Figure 8.** Projected future changes ((RCP4.5-Baseline)/Baseline×100%) in precipitation by the forcing GCM HadGEM2-AO, the MME and each of the five RCMs.

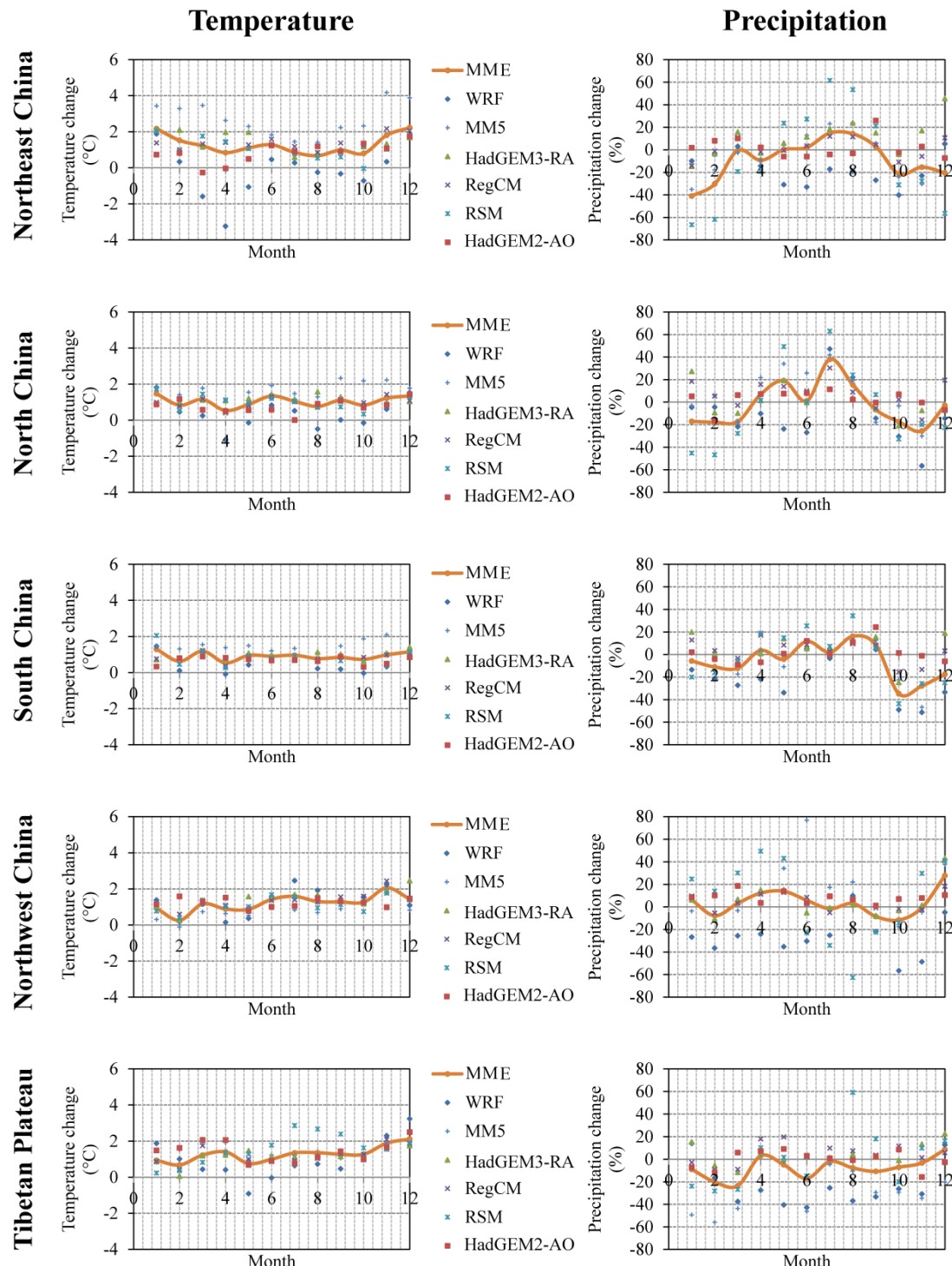

**Figure 9.** Projected future changes in monthly mean temperature and precipitation by the forcing GCM HadGEM2-AO, the MME and each of the five RCMs under RCP4.5 scenario.

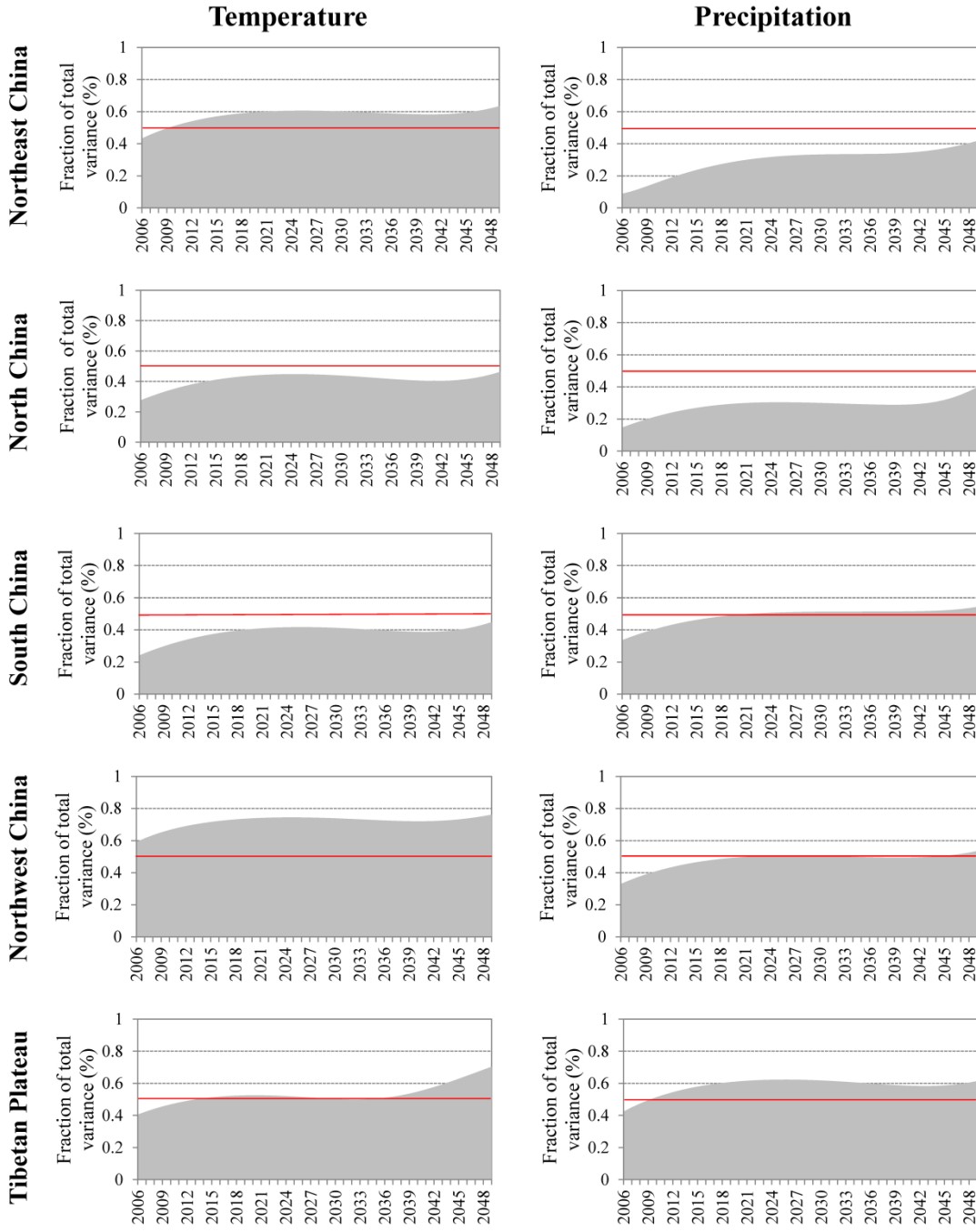

**Figure 10**. Fraction of total variance in future temperature (left panel) and precipitation (right panel) projections explained by intermodel variability (gray) and internal variability (white) over the five subregions.