# Peer review of "High-resolution ensemble projections and uncertainty assessment of regional climate change over China in CORDEX East Asia"

_Hydrology and Earth System Sciences, 2017_

## Referee Comment (RC1) · Anonymous Referee #1 · 1 Oct 2017

This paper reports a useful analysis of model simulations and forecasts of temperature and precipitation over China. Yet the presentation needs improving by avoiding vague and empty statements and the English needs polishing before the paper is publishable.

Section 2, Data and methods lacks details. Why selecting these five RCMs? What advantages do they have compared to other regional and global models products? Do the five models have desired features for the purpose of this analysis? CRU and APHRO products are used as "observations". Are they more accurate and reliable than other global temperature and precipitation data products over the study domain (China)? Section 2.3 is somewhat confusing due to lack of details. Why using Taylor diagram? A

concise description of the Taylor diagram is needed for those who are not familiar with the method. Eqs. (4)-(5) appear to come from nowhere with undefined notations. A justification of the statistical method and metrics used in the analysis is helpful. Section 3 is not well organized and thought out. Overall, discussions are somewhat superficial. To make this paper useful, more insightful explanations and suggestions should be made explicit and specific. For example, on page 6 "All RCMs successfully simulate the precipitation patterns but with quite large biases in amounts". Should we trust more the CRU data or the RCMs simulations? The authors suggest that "the multi-model ensemble outperforms the individual RCM in reproducing the observed spatial pattern of precipitation" (page 6). Would it be possible to obtain the "true" climate by having infinite ensembles? In section 3.3.2, it was suggested that "the seasonal precipitation change in multi-model ensemble has larger magnitude and variability than driving GCM. This phenomenon concerns the significance of the model physics and processes for future climate projection". Specification of what model physics and processes are important would be very useful. The paper ended with "More reliable future climate information could be provided by coupling GCMs and RCMs through the modifications to model structures and parameters." To be specific about the model structures and parameters to be modified would be the valuable new knowledge that the reader can learn from this analysis.

The paper needs a careful text editing to improve its presentation. A long sentence is often confusing such as "Reliable regional future climate projection is important for the evaluation of climate change impacts and vulnerability, as well as the elaboration of appropriate mitigation and adaptation measures, especially for the developing countries like China tend to be one of the most vulnerable to the adverse effects of climate changes" (page 1). English Grammar needs to checked carefully. For example, "The ongoing coordinated regional downscaling experiment (CORDEX) (Giorgi et al., 2009; Jones et al., 2011), whose aim to provide high-resolution regional future climate projections for the majority of populated land regions on the globe by using multi-RCMs, and an interface to the applicants of the climate simulations in climate change impact,

adaptation, and mitigation studies." (page 2) is not a sentence as it does not have a verb.

---

## Referee Comment (RC2) · Anonymous Referee #2 · 30 Oct 2017

Focusing on five sub-regions in China, this study collected five regional climate model simulations from CORDEX-EA, and evaluated their climatology, spatiotemporal variability, and seasonal cycles of the simulated precipitation and temperature. Future projections with uncertainties under RCP4.5 scenario during 2030∼2049 were also analyzed. Generally, this study can provide some insights on dynamical downscaling of regional climate change. However, its novelty is greatly limited partly due to a lack of clarification and a lot of grammar mistakes. Therefore, I suggest a major revision.

Major comments (1) Introduction. The limitation and development of GCMs are reviewed, but the advantages and applications of RCMs are not clearly discussed. A

more detailed introduction on the progress and limitation on dynamical downscaling is needed. As mentioned by the authors, "The CORDEX-EA has been evaluated for simulating the precipitation and temperature over East Asia (Huang et al., 2015; Jin et al., 2016; Lee and Hong, 2014; Oh et al., 2013; Park et al., 2013; Suh et al., 2012; Zou et al., 2014)." Therefore, how does this study differ from previous CORDEX-EA studies should be clearly stated.

(2) Uncertainty quantification method. P5, L5~7. The paper by Hawkins and Sutton (2009, BAMS) used a model-weighted variance when calculating inter-model variability M(t), while eq. 5 in this paper seemed to get a unweighted value. Given that eq. 4 defined a weighted mean of variance as V (same as Hawkins and Sutton's paper), I suggest keeping it consistent in the manuscript, because RCM simulations may differ a lot in both magnitude and variation. If the eq. 5 is just a typo and this study does calculate weights for different models, both simple multi-model ensemble (MME) and weighted MME should be compared in the evaluation (e.g., Figures 2-4).

(3) The abstract needs a careful revision. For example, how does the CORDEX-EA future projection over China or East Asia differ from existing reports (e.g., IPCC AR5 report or at least the driven GCM in this study)? Are the 5 models (RCMs) enough to quantify the model variability? What is the added value for dynamical downscaling (e.g., how much error has been reduced)?

(4) Figure 4b. Why there is a decrease in precipitation correlation, where GCM outperforms all RCMs over North China?

(5) There are a lot of grammar errors while I just mentioned quite a few below. Please proofread the paper carefully or ask a native English speaker for help.

Minor comments (6) P3, Section 2.1. Two datasets were used as reference precipitation, CRU and APHRO. The reason why both datasets are necessary is equivocal, partly because of little comparison between them. Which one was chosen as reference value when calculating precipitation biases (%) in Figure 3 and why? (7) P1, L16,
"decreases -7.8%" -> "decreases by -7.8%". (8) P1, L20, "contribute" -> "contributes". (9) P1, L21, "which" -> "where". (10) P2, L22, "forces on" -> "focusing on". (11) P2, L24-27, this sentence is awkward. (12) P2, L32, "simulating"->"simulation" (13) P3, L2, "will became"->"will become" (14) P3, L13, "Scection 3" ->"Section 3". (15) P4, L1, "include" -> "including", ".. of each of the RCM..." -> "of each RCM ...". (16) Several sentences in the manuscript are difficult to read with grammar mistakes, for instance, P2 L2, P2 L7∼L8, P3 L1, P3 L19∼21, etc. The authors should improve the presentation, especially for Abstract and Introduction Section. (17) Caption of Figure 4 needs revision, where the information for temperature (red rectangles) is missing.

---

## Author Comment (AC1) · 4 Jan 2018

**REPLIES TO THE REVIEWERS' COMMENTS**

The authors are grateful to the reviewers for their valuable comments that helped to improve the quality of the manuscript. The point-by-point responses are presented as follows:

**Reviewer #1**

1.This paper reports a useful analysis of model simulations and forecasts of temperature and precipitation over China. Yet the presentation needs improving by avoiding vague and empty statements and the English needs polishing before the paper is publishable.

Response: Thanks for your comments and suggestions. We tried our best to revise the manuscript according to your advices. Hopefully, this revised version will be satisfactory to meet the publication standard.

2. Section 2, Data and methods lacks details. Why selecting these five RCMs? What advantages do they have compared to other regional and global models products? Do the five models have desired features for the purpose of this analysis?

Response: Thanks for your suggestions. Data and methods in section 2 have been modified in the revision. The reason why five RCMs are selected is below:

The selected five RCMs have been demonstrated to have abilities to reasonably reproduce the regional climate over East Asia and have been used for modeling and predicting extreme climate as well as investigating physical processes of East Asia climate (Cha and Lee, 2009; Cha et al., 2011; Hong and Yhang, 2010; Park et al., 2008; Yhang and Hong, 2008). Moreover, the five RCMs used in this work are derived from the CORDEX East Asia experiment that is able to provide a common framework in a global-wide perspective for regional climate projections in order to understand their uncertainties as well as provide model evaluation.

3. CRU and APHRO products are used as "observations". Are they more accurate and reliable than other global temperature and precipitation data products over the study domain (China)?

Response: Thanks. We use the temperature data from CRU and precipitation data from APHRO as the observation climate in this study. Some illustrations about CRU and APHRO products and the reason why they are used in this study are clarified as below:

Some studies have focused on comparing and evaluating the spatio-temporal similarities and differences of several widely used observed gridded datasets over China (Sun et al., 2014; Wu and Gao, 2013; Yin et al., 2015). Table 1 shows the information of several widely used global observed gridded climate datasets (from Sun et al., 2014). According to Sun et al (2014), all temperature datasets in table 1 exhibit similar distribution patterns for the annual average temperature in mainland China. Considering its easier access and wider usage in evaluation of RCM model used in East Asian/China (Wang et al., 2017), CRU other than

UDEL temperature data are used to evaluate the performance of RCM in this study.

  Table 1 Detailed information on the datasets in the research of Sun et al (2014)

| Dataset | Pre | Tas | Spatial domain | Temporal domain | Reference |
|---|---|---|---|---|---|
| APHRO | √ | | 0.25°, East Asia | Daily, 1951-2007 | (Yatagai et al., 2012) |
| CRU | √ | √ | 0.5°, global | Monthly, 1901-2017 | (New et al., 2000) |
| GPCC | √ | | 0.5°, global | Monthly, 1901-2010 | (Becker et al., 2013) |
| UDEL | √ | √ | 0.5°, global | Monthly, 1901-2010 | (Willmott and Matsuura, 2001) |

Sun et al (2014) suggest that observed precipitation coming from different datasets do have differences, which are caused by differences in raw data sources, quality control schemes, orographic correction and interpolation techniques. Indeed, we have no ability to know the 'truth value'. To some degree, the dataset constructed based on observations from more meteorological stations can be treated as more accurate and reliable one. Among the several precipitation datasets shown in table 1, APHRO's daily gridded precipitation, presently the only long-term, continental-scale, high-resolution daily product, is constructed based on data collected at 5000-12000 stations, which represent 2.3-4.5 times the data made available through the Global Telecommunication System network used for generating global gridded dataset (i.e. CRU, GPCC and UDEL) (Yatagai et al., 2012). Thus, the APHRO

dataset would give more confidence in the robustness of the results in comparison with other global precipitation datasets and thus is widely used for evaluating the performance of RCM

in East Asia (Gao et al., 2017; Kumar and Dimri, 2017; Lau et al., 2017; Lee et al., 2017; Um et al., 2017).

4. Section 2.3 is somewhat confusing due to lack of details. Why using Taylor diagram? A

concise description of the Taylor diagram is needed for those who are not familiar with the method.

Response: Thanks. Detailed illustration for Taylor diagram has been added in the revised manuscript.

The Taylor diagram was designed to quantify the degree of correspondence between the modeled and observed behavior by plotting a 2D graph with three statistics (Pearson correlation coefficient (R), standard deviation (SD), and the root-mean-square error (RMSE)).

In the Taylor diagram, a smaller distance between the observation and the compared models means a closer agreement (Baker and Taylor, 2016; Sun et al., 2015; Taylor, 2001). More details about this diagram are available from the above references. In general, The Taylor diagram enable statistics for different fields (with different units) to show in a single plot, facilitating the comparative assessment of different models.

5. Eqs. (4)-(5) appear to come from nowhere with undefined notations. A justification of the statistical method and metrics used in the analysis is helpful.

Response: Thanks. More details about notations in Eqs. (4)-(5) and methods (where Eqs. (4)-(5) are included) to separate and quantify the two sources of uncertainty were added in the revised manuscript. Here we give a brief illustration.

(1) Firstly, the percentage change from the mean of 1980-1999 is calculated for each projection, and a smooth fourth-order polynomial is fitted for 2030-2049. Then the raw simulation of each model $X_{m,t}$ for the model m and year t which can be expressed by

$$X_{m,t} = x_{m,t} + c_m + \varepsilon_{m,t} \qquad \text{(Eqs. 1)}$$

where the smooth fit is represented by $x_{m,t}$, the reference data is denoted by $c_m$, and the residual is denoted by $\varepsilon_{m,t}$.

**The internal variability** is represented by the decadal mean residuals from these smooth fits for 2030-2049, which is assumed to be constant with lead time.

**The model uncertainty** is considered by the model spread around the mean for each scenario.

(2) The RCMs are weighted by their performance in simulating the current climate from the mean of 1980-1999, up to the year 1999. Thus, each model is weighted according to

$$w_m = \frac{1}{x_{obs} + \left| x_{m,1999} - x_{obs} \right|} \qquad \text{(Eqs. 2)}$$

where $x_{m,1999}$ is the model climate changes at the year of 1999, relative to 1980-1999, and $x_{obs}$ is an observational estimate derived from fitting a similar fourth-order polynomial to the observations. The normalized quantities of these weightings can be expressed as

$$W_m = \frac{w_m}{\sum_m w_m} \qquad \text{(Eqs. 3)}$$

(3) **The internal variability** (equ. 4) is defined as the multi-model mean of theses variance of the residuals from the fits for each model. Here $var_t(.)$ indicates the variance across different time slices.

$$V = \sum_m W_m \, var_t(\varepsilon_m, t) \qquad \text{(Eqs. 4)}$$

(4) **The intermodel variability** (equ.5) is estimated from the weighted variance (var$^w$) in
the different RCM prediction fits ($x_{m,t}$), where var$_m$(.) represents the variance across different
models.

$$M(t) = \mathrm{var}_m^w(x_{m,t}) \qquad \text{(Eqs. 5)}$$

(5) It was assumed that the two sources of uncertainty can be treated independently (i.e.,
there is no interaction between them). Thus, **the total variability** $V_T$ is:

$$V_T(t) = V + M(t) \qquad \text{(Eqs. 6)}$$

6. Section 3 is not well organized and thought out. Overall, discussions are somewhat
superficial. To make this paper useful, more insightful explanations and suggestions should
be made explicit and specific. For example, on page 6 "All RCMs successfully simulate the
precipitation patterns but with quite large biases in amounts". Should we trust more the CRU
data or the RCMs simulations?

Response: Thanks. We reorganized the Section 3 and included more specific analysis in our
revised manuscript. The response to the question "Should we trust more the CRU data or the
RCMs simulations?" is below:

In this paper, we aimed to evaluate the performance of five RCMs within CORDEX-EA
in reproducing present-day climate and to analyze the projected future climate changes under
the middle emission scenario and uncertainties attributed to RCMs and internal variability.
Here the performance of five RCMs in reproducing present-day climate is evaluated by
comparing the RCM simulations with the CRU and APHRO products. The CRU and APHRO
products are constructed based on observed metrological data during historical period. Thus
the CRU and APHRO database can be treated as the proxy for the observed metrological data,
with higher reliability than the RCMs simulations during historical period.

7. The authors suggest that "the multi-model ensemble outperforms the individual RCM in
reproducing the observed spatial pattern of precipitation" (page 6). Would it be possible to
obtain the "true" climate by having infinite ensembles?

Response: Thanks. It is difficult to obtain the "true" climate by having infinite ensembles so
far. The reason is listed below:

The skill of climate models in reproducing precipitation or temperature is limited by
internal atmospheric variability that is largely unpredictable (Kharin and Zwiers, 2002). Thus,
perfect climate model does not exist. Some researchers have concluded the multi-model
ensemble outperforms the individual RCM in reproducing climate pattern (Huttunen et al.,

2017; Rozante et al., 2014). Moreover, the probability of obtaining "true" climate would rise with increased ensemble number. However, huge computational resource is required for the long-term and high-resolution climate projection. Therefore, to obtain the "true" climate by having infinite ensembles is difficult so far.

8. In section 3.3.2, it was suggested that "the seasonal precipitation change in multi-model ensemble has larger magnitude and variability than driving GCM. This phenomenon concerns the significance of the model physics and processes for future climate projection". Specification of what model physics and processes are important would be very useful. The paper ended with "More reliable future climate information could be provided by coupling GCMs and RCMs through the modifications to model structures and parameters." To be specific about the model structures and parameters to be modified would be the valuable new knowledge that the reader can learn from this analysis.

Response: Thanks for your suggestions. The illustrations for important model physics processes have been added in the revision. They are clarified by two points below:

(1) In section 3.3.2, it was suggested that "the seasonal precipitation change in multi-model ensemble has larger magnitude and variability than driving GCM". The configurations of each RCM were showed in Table 2. For each RCM, optimal schemes of the dynamical and physical processes were determined through the investigation of the model sensitivities to the schemes. In general, convective parameterization is the most important and sensitive physical process associated with the simulation results (Huang and Gao, 2017). Land surface parameterizations, as well as those parameterizations over the ocean, are also very important because they control the quantity of moisture entering into atmosphere from the Earth's surface (Zhao and Li, 2015). Thus, the phenomenon above could be attributed to the difference in convective parameterization, land surface parameterizations, as well as those parameterizations over the ocean between GCMs and RCMs. On the other hand, the discrepancies between the RCMs and driving GCM indicate that the RCM projections are sensitive to local and regional processes and the methods represented in the model (Diallo et al., 2012; Saini et al., 2015).

(2) At the end of this paper, further research in the future was added: Further research as for improving the performance of RCM in modeling summer precipitation over South China and the Tibetan Plateau is needed in the future.

**Table 2. RCMs used in this study[a]**

|  | HadGEM3-RA | RegCM4 | MM5 | WRF | RSM |
|---|---|---|---|---|---|
| Resolution | 0.44° | 50km | 50km | 50km | 50km |
| Dynamic process | Non-hydrostatic | Hydrostatic | Non-hydrostatic | Non-hydrostatic | Hydrostatic |

| Convective scheme | Revised mass flux scheme | MIT-Emanuel | Kain-Fritch II | Kain-Fritch II | Simplified Arakawa-Schubert |
|---|---|---|---|---|---|
| Land surface parameterization | MOSES2 | CLM3 | CLM3 | NOAH | NOAH |
| Planetary boundary layer | MOSES2 non-local | Holtslag | YSU | YSU | YSU |
| Spectral nudging | No | Yes | Yes | Yes | Yes |
| Center of research | MOHC | ICTP | NCAR | NCAR | YSU |
| References | Davies et al.(2005) | Giorgi et al.(2012) | Cha and Lee(2009) | Skamarock et al.(2005) | Hong et al.(2013) |

[a]MOSES= Met Office Surface Exchange Scheme, CLM= Community Land Model, NOAH=Noah Land Surface Model, YSU= Yonsei University scheme, MOHC= The Met Office Hadley Centre, ICTP= The International Centre for Theoretical Physics, NCAR= National Center for Atmospheric Research

9.The paper needs a careful text editing to improve its presentation. A long sentence is often confusing such as "Reliable regional future climate projection is important for the evaluation of climate change impacts and vulnerability, as well as the elaboration of appropriate mitigation and adaptation measures, especially for the developing countries like China tend to be one of the most vulnerable to the adverse effects of climate changes" (page 1). English Grammar needs to checked carefully. For example, "The ongoing coordinated regional downscaling experiment (CORDEX) (Giorgi et al., 2009; Jones et al., 2011), whose aim to provide high-resolution regional future climate projections for the majority of populated land regions on the globe by using multi-RCMs, and an interface to the applicants of the climate simulations in climate change impact, adaptation, and mitigation studies." (page 2) is not a sentence as it does not have a verb.

Response: Sorry for the serious language problem in previous manuscript. We consider your criticism thoroughly in revising manuscript. In total, the previous article was severely revised four times, particularly on the presentation, interpretation and language together with the figures and tables. In the revising process, two important co-authors (Prof. W. R. Peltier from University of Toronto, Toronto, Canada and Prof. Guiling Wang from University of Connecticut, USA) with proficient English skills contributed to the thorough control check in language for this version significantly. They read and corrected the language and presentation for the paper sentence by sentence to meet the reviewers' request. As you can see from the track-changes in the main context, tables, and figures, the revised version was really undergone a major revision through which the paper quality has been improved.

**Reviewer #2**

**Major comments**

(1) Introduction. The limitation and development of GCMs are reviewed, but the advantages and applications of RCMs are not clearly discussed. A more detailed introduction on the progress and limitation on dynamical downscaling is needed. As mentioned by the authors, "The CORDEX-EA has been evaluated for simulating the precipitation and temperature over East Asia (Huang et al., 2015; Jin et al., 2016; Lee and Hong, 2014; Oh et al., 2013; Park et al., 2013; Suh et al., 2012; Zou et al., 2014)." Therefore, how does this study differ from previous CORDEX-EA studies should be clearly stated.

Response: Thanks for your valuable suggestions. More details on the progress and limitation on dynamical downscaling and the difference between this study and previous CORDEX-EA studies were added in the revision. Two points are clarified as follows:

(1) The resolution of RCMs is approximately 12-50 km, and it accounts for the sub-GCM grid-scale forcing, e.g. complex topographical features and land cover heterogeneities in a physically based manner. However, RCMs inherit the biases from systematic model errors caused by imperfect conceptualization, discretization, and spatial averaging within grid cells. (Dong et al., 2018). Nonetheless, RCM ensembles enable the understanding and characterization of uncertainties which have different origins, from the future scenario, to the forcing data and the regional model physics, and therefore, reduce uncertainties and increase confidence in future projections.

(2) A series of studies based on RCMs within CORDEX-EA have been conducted to project extreme and mean precipitation and temperature over china under different scenarios (Niu et al., 2015; Jin et al., 2016; Lee et al., 2014; Park et al., 2016; Tang et al., 2016; Um et al., 2017), but little attention has been paid to quantify the contributions of the uncertainty arising from RCMs and internal variability in future climate projection over China. Thus, it is necessary to objectively evaluate the capability of RCMs and quantify the uncertainty in future climate projections. In this study, we evaluate the performance of five RCMs within CORDEX-EA to reproduce present-day climate and to analyze the projected future climate changes under the middle emission scenario. More importantly, biases in current climate simulations and uncertainties in future climate projections attributed to the RCMs and internal variability are further analyzed.

(2) Uncertainty quantification method. P5, L5-7. The paper by Hawkins and Sutton (2009, BAMS) used a model-weighted variance when calculating inter-model variability $M(t)$, while eq. 5 in this paper seemed to get an unweighted value. Given that eq. 4 defined a weighted mean of variance as V (same as Hawkins and Sutton's paper), I suggest keeping it consistent in the manuscript, because RCM simulations may differ a lot in both magnitude and variation.

If the eq. 5 is just a typo and this study does calculate weights for different models, both
simple multi-model ensemble (MME) and weighted MME should be compared in the
evaluation (e.g., Figures 2-4).

Response: Thanks for your valuable suggestions. Equation 5 was modified and the weighted
variance was used when calculating the inter-model variability in the revision. As shown in
the Figures 1-3 in this response file, no significant difference in the spatial patterns (Figures
1-2) between simple multi-model ensemble (MME) and weighted MME can be found.
Similarly, skills of the models in reproducing the precipitation and temperature with simple
MME are nearly consistent with that based on weighted MME (Figure 3). Thus, the weighted
MME is used in the revised manuscript, instead of the simple MME.

[Figure]

Figure 1. Spatial distributions of annual average temperature (ºC) of CRU (a), multi-model
ensemble (b), multi-model ensemble (c), and temperature biases (ºC) of the driving GCM
HadGEM2-AO (d), multi-RCM ensemble (e, f) and five RCMs (g-k) during 1980-2005.

[Figure]

Figure 2. Spatial distributions of annual average precipitation (mm/year) of APHRO (a), multi-model ensemble (b), weighted multi-model ensemble (c), and precipitation biases (%) of the driving GCM HadGEM2-AO (d), multi-RCM ensemble (e and f) and five RCMs (g-k) during 1980-2005.

[Figure]

Figure 3. Taylor diagram to compare the skill of the models in representing the annual average temperature and precipitation over the five regions of China, using the CRU (for temperature) and APHRO (for precipitation) data as the REF.

(3) The abstract needs a careful revision. For example, how does the CORDEX-EA future projection over China or East Asia differ from existing reports (e.g., IPCC AR5 report or at least the driven GCM in this study)? Are the 5 models (RCMs) enough to quantify the model variability? What is the added value for dynamical downscaling (e.g., how much error has been reduced)?

Response: Thanks for your suggestion. We tried to compare and add the CORDEX-EA future projection and the simulation by the driven GCM in the revision. Meanwhile, the added value for dynamical downscaling was analyzed in the revised manuscript.

(1) The comparison of the CORDEX-EA future projection over China with the projection by the driven GCM was added. As shown in table 3, increases in annual mean temperature based on the five RCMs' ensemble range from 0.9 ℃ to 1.3 ℃ in different subregions, which is quite close to the projected increase in annual mean temperature from the forcing GCM (range from 0.7 ℃ to 1.4 ℃). Meanwhile, similar spatial patterns for projected change in annual mean temperature by the ensemble method and the driving GCM are shown in Figures 4a-b. Generally, the CORDEX-EA future projected change in mean temperature is nearly consistent with the results from the driving GCM. However, opposite signals for projected changes in average precipitation between the ensemble method and the driving GCM are shown over South china, Northeast china and Tibetan Plateau (table 3). Particularly the spatial and temporal differences in projection from two methods above are largest at the Tibetan Plateau, up to about 10%.

Table 3. The future changes in average temperature (T; °C) and precipitation (P; %) for the five subregions (as shown in Figure 1). The ensemble averages for each statistic are given in the second line. The projections by the forcing GCM are given in the last line.

| | | WRF | MM5 | HadGEM3-RA | RegCM | RSM | Ensemble | HadGEM2-AO |
|---|---|---|---|---|---|---|---|---|
| Northeast China | T(°C) | 0.2 | 2.7 | 1.4 | 1.4 | 1.1 | 1.3 | 0.8 |
| | P(%) | -21.7 | 8.2 | 13.0 | 4.4 | 7.1 | 1.5 | -0.4 |
| North China | T(°C) | 0.3 | 1.7 | 1.1 | 1.0 | 1.0 | 1.0 | 0.8 |
| | P(%) | -1.5 | 15.1 | 3.1 | 10.2 | 3.3 | 6.1 | 4.9 |
| South China | T(°C) | 0.5 | 1.5 | 1.0 | 0.8 | 0.8 | 0.9 | 0.7 |
| | P(%) | -14.6 | -1.6 | 4.8 | 4.9 | 1.3 | -1.5 | 2.3 |
| Northwest China | T(°C) | 1.3 | 0.8 | 1.5 | 1.3 | 1.1 | 1.2 | 1.2 |
| | P(%) | -27.0 | 19.4 | 2.2 | 4.7 | 8.9 | 3.6 | 7.2 |
| Tibetan Plateau | T(°C) | 0.9 | 1.4 | 1.2 | 1.3 | 1.6 | 1.3 | 1.4 |
| | P(%) | -31.6 | -17.8 | 2.4 | 6.4 | 7.4 | -7.8 | 2.1 |

(2) The added values for RCMs were confirmed by comparing the performance of RCM

and GCM in reproducing annual mean precipitation and temperature during historical period.

According to the Taylor diagram (Figure 3 above), it is found that the added value for RCMs strongly depends on the climate variable and the region of interest. The added value of the

RCMs with respect to the driving global climate model was evident in term of annual mean temperature over all five subregions, with higher spatial correlation coefficient for all five

RCMs. Compared with the driving global climate model simulations, the spatial patterns of the simulated annual average precipitation over South China, Northwest China and the

Tibetan Plateau were improved in most RCMs. The expectations are over Northeast China and North China, where higher performance is shown for the driving global climate model.

Please see lines 288-299 in this response file for the reasons resulting in this phenomenon.

Besides, the results shown in above two points were summarized in a couple of sentences in the revised abstract, in view of the length limit for the abstract.

[Figure]

Figure 4. Projected future changes (RCP4.5-Baseline) in surface air temperature for the forcing GCM HadGEM2-AO and each of the five RCMs.

[Figure]

Figure 5. Projected future changes ((RCP4.5-Baseline)/Baseline×100%) in precipitation for the forcing GCM HadGEM2-AO and each of the five RCMs.

(4) Figure 4b. Why there is a decrease in precipitation correlation, where GCM outperforms all RCMs over North China?

Response: Thanks. The reason why there is a decrease in precipitation correlation over North China was added in the revision. In this study, it is found the performance of RCM in reproducing spatial pattern of annual average precipitation is superior to that of the driving GCM in term of correlation coefficient in most sub-regions over China. The only exception is North China. In reality, the added value in RCM simulations (in compaction with GCM) is related to a better representation of spatial variability of surface climate statistics, particularly in regions with fine-scale surface forcing such as orographic and coastal features. Thus, the added value in RCM simulations is commonly significant in regions with fine-scale surface forcing, whereas the performance of RCM is less improved or even worse than that of the driving GCM over relatively flat regions. For instance, Prommel and Geyer (Prömmel et al.,

2010) also found the RCM deteriorates some results compared to the driving GCM in
relatively flat subregions surrounding the Alps, particularly during the summer season.

(5) There are a lot of grammar errors while I just mentioned quite a few below. Please
proofread the paper carefully or ask a native English speaker for help.

Response: Sorry for the serious language problem in previous manuscript. In the revising
process, two important co-authors (Prof. W. R. Peltier from University of Toronto, Toronto,
Canada and Prof. Guiling Wang from University of Connecticut, USA) with proficient
English skills contributed to the thorough control check in language for this version
significantly. As you can see from the track-changes in the main context, tables, and figures,
the revised version was really undergone a major revision through which the paper quality
has been improved.

**Minor comments**

(6) P3, Section 2.1. Two datasets were used as reference precipitation, CRU and APHRO.
The reason why both datasets are necessary is equivocal, partly because of little comparison
between them. Which one was chosen as reference value when calculating precipitation
biases (%) in Figure 3 and why?

Response: Thanks for your suggestions. In figure 3 APHRO data was chosen as reference
precipitation when calculating precipitation biases (%). Meanwhile, only APHRO dataset
other than CRU dataset was used as reference precipitation in the revision, to increase the
readability of this paper. The reason why APHRO dataset is used has been detailed in lines
40-53 in this response file.

(7) P1, L16, "decreases -7.8%" -> "decreases by -7.8%".

Response: Thanks. They have been done.

(8) P1, L20, "contribute" -> "contributes".

Response: Thanks. They have been done.

(9) P1, L21, "which" -> "where".

Response: Thanks. They have been done.

(10) P2, L22, "forces on" -> "focusing on".

Response: Thanks. They have been done.

(11) P2, L24-27, this sentence is awkward.

Response: Thanks. We rewrote this sentence.

(12) P2, L32, "simulating"->"simulation"

Response: Thanks. They have been done.

(13) P3, L2, "will became"->"will become"

Response: Thanks. They have been done.

(14) P3, L13, "Scection 3" ->"Section 3".

Response: Thanks. They have been done.

(15) P4, L1, "include" -> "including", ".. of each of the RCM: : :" -> "of each RCM : : :".

Response: Thanks. They have been done.

(16) Several sentences in the manuscript are difficult to read with grammar mistakes, for
instance, P2 L2, P2 L7-L8, P3 L1, P3 L19-21, etc. The authors should improve the
presentation, especially for Abstract and Introduction Section.

Response: Thanks. We rewrote these sentences.

(17) Caption of Figure 4 needs revision, where the information for temperature (red
rectangles) is missing.

Response: Thanks. We modified this caption in the revised manuscript.

**References:**

Baker, N.C. and Taylor, P.C.: A Framework for Evaluating Climate Model Performance Metrics. Journal of Climate, 29, 1773-1782, doi: 10.1175/JCLI-D-15-0114.1, 2016

Becker, A., Finger, P. and Meyer-Christoffer, A. et al.: A description of the global land-surface precipitation data products of the Global Precipitation Climatology Centre with sample applications including centennial (trend) analysis from 1901鈥損resent. Earth System Science Data, 5, 71-99, doi: 10.5194/essd-5-71-2013, 2013

Cha, D. and Lee, D.: Reduction of systematic errors in regional climate simulations of the summer monsoon over East Asia and the western North Pacific by applying the spectral nudging technique. Journal of Geophysical Research: Atmospheres, 114, D14108, doi: 10.1029/2008JD011176, 2009

Cha, D., Jin, C. and Lee, D. et al.: Impact of intermittent spectral nudging on regional climate simulation using Weather Research and Forecasting model. Journal of Geophysical Research: Atmospheres, 116, D10103, doi: 10.1029/2010JD015069, 2011

Davies, T., Cullen, M.J.P. and Malcolm, A.J. et al.: A new dynamical core for the Met Office's global and regional modelling of the atmosphere. Quarterly Journal of the Royal Meteorological Society, 131, 1759-1782, doi: 10.1256/qj.04.101, 2005

Diallo, I., Sylla, M.B. and Giorgi, F. et al.: Multimodel GCM-RCM Ensemble-Based Projections of Temperature and Precipitation over West Africa for the Early 21st Century. International Journal of Geophysics, 2012, Article ID 972896, doi: 10.1155/2012/972896, 2012

Dong, N.D., Jayakumar, K.V. and Agilan, V.: Impact of Climate Change on Flood Frequency of the Trian Reservoir in Vietnam Using RCMS. Journal of Hydrologic Engineering, 23, 05017032, doi: 10.1061/(ASCE)HE.1943-5584.0001609, 2018

Gao, J., Hou, W. and Xue, Y. et al.: Validating the dynamic downscaling ability of WRF for East Asian summer climate. Theoretical and Applied Climatology, 128, 241-253, doi: 10.1007/s00704-015-1710-9, 2017

Giorgi, F., Coppola, E. and Solmon, F. et al.: RegCM4: model description and preliminary tests over multiple CORDEX domains. Climate Research, 52, 7-29, doi: 10.3354/cr01018, 2012

Hong, S. and Yhang, Y.: Implications of a Decadal Climate Shift over East Asia in Winter: A Modeling Study. Journal of Climate, 23, 4989-5001, doi: 10.1175/2010JCLI3637.1, 2010

Hong, S., Park, H. and Cheong, H. et al.: The Global/Regional Integrated Model system (GRIMs). Asia-Pacific Journal of Atmospheric Sciences, 49, 219-243, doi: 10.1007/s13143-013-0023-0, 2013

Huang, D. and Gao, S.: Impact of different cumulus convective parameterization schemes on the simulation of precipitation over China. Tellus A: Dynamic Meteorology and Oceanography, 69, 1406264, doi: 10.1080/16000870.2017.1406264, 2017

Huttunen, J.M.J., Räisänen, J. and Nissinen, A. et al.: Cross-validation analysis of bias models in Bayesian multi-model projections of climate. Climate Dynamics, 48, 1555-1570, doi: 10.1007/s00382-016-3160-1, 2017

Jin, C., Cha, D. and Lee, D. et al.: Evaluation of climatological tropical cyclone activity over the western North Pacific in the CORDEX-East Asia multi-RCM simulations. Climate Dynamics, 47, 765-778, doi: 10.1007/s00382-015-2869-6, 2016

Kharin, V.V. and Zwiers, F.W.: Climate Predictions with Multimodel Ensembles. Journal of Climate,
15, 793-799, doi: 10.1175/1520-0442(2002)015<0793:CPWME>2.0.CO;2, 2002

Kumar, D. and Dimri, A.P.: Regional climate projections for Northeast India: an appraisal from
CORDEX South Asia experiment. Theoretical and Applied Climatology, doi:
10.1007/s00704-017-2318-z, 2017

Lau, W.K.M., Kim, K. and Ruby Leung, L.: Changing circulation structure and precipitation
characteristics in Asian monsoon regions: greenhouse warming vs. aerosol effects. Geoscience
Letters, 4, 28, doi: 10.1186/s40562-017-0094-3, 2017

Lee, D., Min, S. and Jin, J. et al.: Thermodynamic and dynamic contributions to future changes in
summer precipitation over Northeast Asia and Korea: a multi-RCM study. Climate Dynamics, 49,
4121-4139, doi: 10.1007/s00382-017-3566-4, 2017

Lee, J., Hong, S. and Chang, E. et al.: Assessment of future climate change over East Asia due to the
RCP scenarios downscaled by GRIMs-RMP. Climate Dynamics, 42, 733-747, doi:
10.1007/s00382-013-1841-6, 2014

New, M.G., Hulme, M. and Jones, P.D.: Representing Twentieth-Century Space–Time Climate
Variability. Part II: Development of 1901–96 Monthly Grids of Terrestrial Surface Climate.
Journal of Climate, 13, 2217-2238, doi: 10.1175/1520-0442(2000)013<2217:RTCSTC>2.0.CO;2,
2000

Niu, X., Wang, S. and Tang, J. et al.: Multimodel ensemble projection of precipitation in eastern China
under A1B emission scenario. Journal of Geophysical Research: Atmospheres, 120, 9965-9980,
doi: 10.1002/2015JD023853, 2015

Park, C., Min, S. and Lee, D. et al.: Evaluation of multiple regional climate models for summer climate
extremes over East Asia. Climate Dynamics, 46, 2469–2486, doi: 10.1007/s00382-015-2713-z,
2016

Park, E.H., Hong, S.Y. and Kang, H.S.: Characteristics of an East–Asian summer monsoon
climatology simulated by the RegCM3. Meteorology and Atmospheric Physics, 100, 139-158, doi:
10.1007/s00703-008-0300-0, 2008

Prömmel, K., Geyer, B. and Jones, J.M. et al.: Evaluation of the skill and added value of a
reanalysis-driven regional simulation for Alpine temperature. International Journal of Climatology,
30, 760-773, doi: 10.1002/joc.1916, 2010

Rozante, J.R., Moreira, D.S. and Godoy, R.C.M. et al.: Multi-model ensemble: technique and
validation. Geoscientific Model Development, 7, 2333-2343, doi: 10.5194/gmd-7-2333-2014,
2014

Saini, R., Wang, G. and Yu, M. et al.: Comparison of RCM and GCM projections of boreal summer
precipitation over Africa. Journal of Geophysical Research: Atmospheres, 120, 3679-3699, doi:
10.1002/2014JD022599, 2015

Skamarock, W.C., Klemp, J.B. and Dudhia, J. et al.: A Description of the Advanced Research WRF
Version 2, 2005.

Sun, Q., Miao, C. and Duan, Q. et al.: Would the 'real' observed dataset stand up? A critical
examination of eight observed gridded climate datasets for China. Environmental Research Letters,
9, 015001, doi: 10.1088/1748-9326/9/1/015001, 2014

Sun, Q., Miao, C. and Duan, Q.: Projected changes in temperature and precipitation in ten river basins
over China in 21st century. International Journal of Climatology, 35, 1125-1141, doi:
10.1002/joc.4043, 2015

Tang, J., Li, Q. and Wang, S. et al.: Building Asian climate change scenario by multi-regional climate
models ensemble. Part I: surface air temperature. International Journal of Climatology, 36,
4241-4252, doi: 10.1002/joc.4628, 2016

Taylor, K.E.: Summarizing multiple aspects of model performance in a single diagram. Journal of
Geophysical Research: Atmospheres, 106, 7183-7192, doi: 10.1029/2000JD900719, 2001

Um, M., Kim, Y. and Kim, J.: Evaluating historical drought characteristics simulated in CORDEX East
Asia against observations. International Journal of Climatology, 37, 4643-4655, doi:
10.1002/joc.5112, 2017

Wang, L., Chen, W. and Huang, G. et al.: Changes of the transitional climate zone in East Asia: past
and future. Climate Dynamics, 49, 1463-1477, doi: 10.1007/s00382-016-3400-4, 2017

Willmott, C.J. and Matsuura, K.: Terrestrial Air Temperature and Precipitation: Monthly and Annual
Time Series (1950 - 1999), 2001.

Wu, J. and Gao, X.: A gridded daily observation dataset over China region and comparison with the
other datasets. Chinese Journal of Geophysics. (in Chinese), 56, 1102-1111, doi:
10.6038/cjg20130406, 2013

Xie, P., Yatagai, A. and Chen, M. et al.: A Gauge-Based Analysis of Daily Precipitation over East Asia.
Journal of Hydrometeorology, 8, 607-626, doi: 10.1175/JHM583.1, 2007

Yatagai, A., Kamiguchi, K. and Arakawa, O. et al.: APHRODITE: Constructing a Long-Term Daily
Gridded Precipitation Dataset for Asia Based on a Dense Network of Rain Gauges. Bulletin of the
American Meteorological Society, 93, 1401-1415, doi: 10.1175/BAMS-D-11-00122.1, 2012

Yhang, Y. and Hong, S.: Improved Physical Processes in a Regional Climate Model and Their Impact
on the Simulated Summer Monsoon Circulations over East Asia. Journal of Climate, 21, 963-979,
doi: 10.1175/2007JCLI1694.1, 2008

Yin, H., Donat, M.G. and Alexander, L.V. et al.: Multi-dataset comparison of gridded observed
temperature and precipitation extremes over China. International Journal of Climatology, 35,
2809-2827, doi: 10.1002/joc.4174, 2015

Zhao, W. and Li, A.: A Review on Land Surface Processes Modelling over Complex Terrain.
Advances in Meteorology, 2015, Article ID 607181, doi: 10.1155/2015/607181, 2015